# FinBen: A Holistic Financial Benchmark for Large Language Models

Qianqian Xie[b,a], Weiguang Han[b], Zhengyu Chen[b], Ruoyu Xiang[a], Xiao Zhang[a], Yueru He[a],
Mengxi Xiao[b], Dong Li[b], Yongfu Dai[g], Duanyu Feng[g], Yijing Xu[a], Haoqiang Kang[e],
Ziyan Kuang[l], Chenhan Yuan[c], Kailai Yang[c], Zheheng Luo[c], Tianlin Zhang[c],
Zhiwei Liu[c], Guojun Xiong[j], Zhiyang Deng[i], Yuechen Jiang[i], Zhiyuan Yao[i],
Haohang Li[i], Yangyang Yu[i,*], Gang Hu[h], Jiajia Huang[k], Xiao-Yang Liu[e,*],
Alejandro Lopez-Lira[d,*], Benyou Wang[f], Yanzhao Lai[m], Hao Wang[g], Min Peng[b,*],
Sophia Ananiadou[c,*], Jimin Huang[a,*]

[a]The Fin AI, [b]Wuhan University, [c]The University of Manchester, [d]University of Florida,
[e]Columbia University, [f]The Chinese University of Hong Kong, Shenzhen,
[g]Sichuan University, [h]Yunnan University, [i]Stevens Institute of Technology
[j]Stony Brook University, [k]Nanjing Audit University,
[l]Jiangxi Normal University, [m]Southwest Jiaotong University

## Abstract

LLMs have transformed NLP and shown promise in various fields, yet their potential in finance is underexplored due to a lack of comprehensive benchmarks, the rapid development of LLMs, and the complexity of financial tasks. In this paper, we introduce FinBen, the first extensive open-source evaluation benchmark, including 42 datasets spanning 24 financial tasks, covering eight critical aspects: information extraction (IE), textual analysis, question answering (QA), text generation, risk management, forecasting, decision-making, and bilingual (English and Spanish). FinBen offers several key innovations: a broader range of tasks and datasets, the first evaluation of stock trading, novel agent and Retrieval-Augmented Generation (RAG) evaluation, and two novel datasets for regulations and stock trading. Our evaluation of 21 representative LLMs, including GPT-4, ChatGPT, and the latest Gemini, reveals several key findings: While LLMs excel in IE and textual analysis, they struggle with advanced reasoning and complex tasks like text generation and forecasting. GPT-4 excels in IE and stock trading, while Gemini is better at text generation and forecasting. Instruction-tuned LLMs improve textual analysis but offer limited benefits for complex tasks such as QA. FinBen has been used to host the first financial LLMs shared task at the FinNLP-AgentScen workshop during IJCAI-2024, attracting 12 teams. Their novel solutions outperformed GPT-4, showcasing FinBen's potential to drive innovations in financial LLMs. All datasets and code are publicly available for the research community[2], with results shared and updated regularly on the Open Financial LLM Leaderboard[3].

---

[*]Corresponding Authors

[2]https://github.com/The-FinAI/PIXIU

[3]Now under the umbrella of FINOS at Linux Foundation, https://finosfoundation/
Open-Financial-LLM-Leaderboard

Table 1: Comparison of different financial benchmarks based on the number of tasks and datasets and the task counts across aspects: information extraction (IE), textual analysis (TA), question answering (QA), text generation (TG), risk management (RM), forecasting (FO), decision-making (DM), and spanish (SP).

| Benchmark | Language | Dataset | Task | IE | TA | QA | TG | RM | FO | DM | SP |
|---|---|---|---|---|---|---|---|---|---|---|---|
| CFBenchmark | Chinese | 8 | 7 | 1 | 3 | ✗ | 3 | ✗ | ✗ | ✗ | ✗ |
| Fin-Eva | Chinese | 1 | 1 | ✗ | ✗ | 1 | ✗ | ✗ | ✗ | ✗ | ✗ |
| PIXIU | English | 15 | 8 | 1 | 3 | 1 | 1 | 1 | 1 | ✗ | ✗ |
| FinanceBench | English | 1 | 1 | ✗ | ✗ | 1 | ✗ | ✗ | ✗ | ✗ | ✗ |
| BizBench | English | 8 | 5 | 2 | ✗ | 2 | 1 | ✗ | ✗ | ✗ | ✗ |
| FinBen | English, Spanish | 42 | 24 | 6 | 8 | 3 | 1 | 4 | 1 | 1 | 4 |

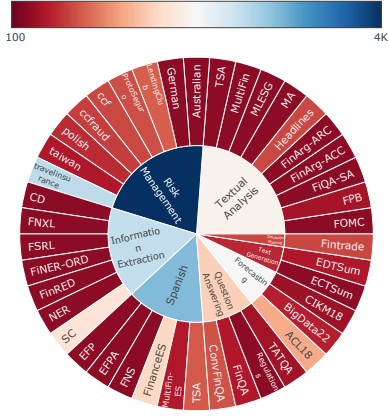

Figure 1: FinBen's evaluation datasets with sizes ranging from 100 to 4,000.

# 1  Introduction

Recently, Large Language Models (LLMs) (Brown et al., 2020) such as ChatGPT[4] and GPT-4 (OpenAI, 2023), have reshaped the field of natural language processing (NLP) and exhibited remarkable capabilities in specialized domains across mathematics, coding, medicine, law, and finance (Bubeck et al., 2023). Within the financial domain, recent several studies (Xie et al., 2023a; Lopez-Lira and Tang, 2023; Li et al., 2023c; Xie et al., 2023b; Liu et al., 2023a; Yang et al., 2023a; Xie et al., 2024) have shown the great potential of LLMs such as GPT-4 on financial text analysis and prediction tasks. While their potential is evident, a comprehensive understanding of their capabilities and limitations for finance remains largely unexplored. This is due to a lack of extensive evaluation studies and benchmarks, and the inherent complexities associated with the professional nature of financial tasks.

Existing financial domain evaluation benchmarks, including PIXIU (Xie et al., 2023b), FinanceBench (Islam et al., 2023) and BizBench (Koncel-Kedziorski et al., 2023), have **limited evaluation tasks** and primarily **focus on financial NLP tasks**, as shown in Table 1. Most existing benchmarks cover only a small number of evaluation tasks and are centered on NLP capabilities, such as information extraction (IE) and question answering (QA) (Huang et al., 2024; Liu et al., 2024b; Hu et al., 2024; Yang et al., 2024; Zhao et al., 2024a,c). While PIXIU stands out by covering the highest number of tasks, it includes only one evaluation task in most categories. This narrow focus limits their ability to comprehensively evaluate LLMs across the diverse and complex landscape of financial applications, such as forecasting, risk management, and decision-making. It is insufficient for a thorough evaluation of LLM capabilities, especially in the financial area.

To bridge this gap, we propose FinBen, a novel comprehensive open-source evaluation benchmark developed through the collaborative efforts of experts in both computer science and finance. As shown in Figure 1, FinBen comprises 42 datasets spanning 24 financial tasks, meticulously organized to assess LLMs across eight critical aspects: information extraction (IE), textual analysis (TA), question answering (QA), text generation (TG), risk management (RM), forecasting (FO), decision-making (DM), and bilingual (English and Spanish). Each category targets specific skills of financial data processing and analysis, ensuring a thorough evaluation of LLMs and showcasing their proficiency in managing complex financial scenarios.

FinBen introduces several innovations over existing benchmarks: 1) **New tasks**: FinBen introduces a significantly larger number of tasks and datasets, making it the most holistic benchmark for financial LLMs with the highest number of tasks and datasets. This extensive range provides a more robust evaluation of LLM capabilities in diverse financial contexts. 2) **Broader coverage**: Covering eight aspects of the financial sector, FinBen is the first benchmark to include the evaluation of stock trading, which is the fundamental task in the financial sector, involving complex decision-making processes that impact market dynamics and investment strategies. 3) **New evaluation strategy**: FinBen is the first benchmark to include agent-based evaluation and retrieval-augmented generation (RAG)

---

[4]https://openai.com/chatgpt

based evaluation. These innovative strategies provide a more dynamic and realistic assessment of LLMs, reflecting their ability to interact with and retrieve relevant information from vast datasets. 4) **Novel datasets**: FinBen proposes two novel open-source datasets of QA and stock trading tasks for the research community, pushing the boundaries of what LLMs can achieve and setting a new standard for dataset comprehensiveness. 5) **Empowering financial LLMs research**: Leveraging financial tasks in FinBen, we hosted the first shared task (see Appendix G for details) focused on financial LLMs at the FinNLP-AgentScen workshop during IJCAI-2024 [5]. This event attracted 12 teams, leveraging our benchmark to develop novel LLMs-based solutions within the financial domain. Remarkably, the proposed methods achieved superior performance compared to GPT-4, demonstrating the benchmark's potential to foster innovations and advance the state-of-the-art (SOTA) in financial LLMs.

Based on FinBen, we assess 21 representative general LLMs such as GPT-4, ChatGPT, and the latest Gemini, and financial LLMs, and have the following findings: 1) **Superior Capabilities with Limitations**: While LLMs exhibit exceptional prowess in IE and textual analysis tasks, they underperform in areas necessitating advanced reasoning and complex IE, such as text generation and forecasting. 2) **Potential in Stock Trading**: SOTA LLMs have demonstrated considerable promise in stock trading applications. However, there remains significant room for improvement due to their limitations in reasoning and comprehensive forecasting abilities. 3) **Closed-Source Superiority**: Closed-source commercial LLMs continue to lead in performance within the financial domain. Specifically, GPT-4 excels in IE, text analysis, QA, and intricate stock trading tasks, while Gemini shows superior capabilities in text generation and forecasting. 4) **Open-Source Improvements and Limitations**: While open-source, instruction-tuned financial LLMs have shown notable enhancements in textual analysis and IE tasks, the advantages of instruction-tuning are less pronounced when it comes to complex tasks such as QA, text generation, and forecasting.

In summary, the main contributions of this paper are: 1) we present FinBen, the first comprehensive open-sourced evaluation benchmark for LLMs in the financial domain, 2) we utilize a novel taxonomy covering eight aspects for organizing financial evaluation tasks, 3) we develop two novel evaluation datasets for the research community, and 4) we conduct systematic evaluation of 21 LLMs using FinBen, showcasing their advantages and limitations and highlighting directions for future work.

## 2 FinBen

In this section, we delve into the specifics of FinBen, detailing the evaluation taxonomy, data sources, and evaluation tasks.

### 2.1 The Taxonomy of Financial Evaluation Tasks

In the dynamic landscape of financial technology, evaluating the capabilities of LLMs necessitates a comprehensive and structured approach. We propose a novel taxonomy for financial evaluation tasks, categorizing and assessing LLMs across eight financial domains inspired by established taxonomies in financial tasks (Cao, 2022; Li et al., 2023b; Zhao et al., 2024b): **Information Extraction (IE)**, **Textual Analysis (TA)**, **Question Answering (QA)**, **Text Generation (TG)**, **Risk Management (RM)**, **Forecasting (FO)**, **Decision-Making (DM)**, and **Spanish (SP)**. **Information Extraction** focuses on identifying key entities and relationships within financial documents, transforming unstructured data into structured insights (Costantino and Coletti, 2008). **Textual Analysis** delves into content and sentiment analysis of financial texts, aiding in market trend understanding (Loughran and McDonald, 2020). **Question Answering** evaluates the model's ability to comprehend and respond to financial queries (Maia et al., 2018). **Text Generation** assesses the production of coherent financial text (La Quatra and Cagliero, 2020). **Risk Management** involves evaluating creditworthiness, detecting fraud, and ensuring regulatory compliance (Aziz and Dowling, 2019). **Forecasting** predicts future financial trends, enabling strategic responses to market dynamics (Abu-Mostafa and Atiya, 1996). **Decision-Making** assesses the model's proficiency in making informed financial decisions, such as developing trading strategies and optimizing investment portfolios (Paiva et al., 2019). Finally, **Spanish** evaluates the model's capabilities in other languages except for English, particularly in low-resource languages.

---

[5] https://sites.google.com/nlg.csie.ntu.edu.tw/finnlp-agentscen

## 2.2 Data Sources

FinBen's evaluation tasks are drawn from three primary data sources: 1) open-sourced datasets from existing studies originally released for non-LLM evaluation settings. Domain experts have designed diverse prompts and reformulated these datasets into instruction-response pairs, making them suitable for evaluating the zero-shot performance of LLMs. 2) datasets from existing evaluation benchmarks such as PIXIU. These datasets have already been transformed into the instruction tuning format, allowing for seamless integration and direct use in FinBen. 3) novel datasets introduced in this paper. These datasets are designed to address gaps in existing benchmarks and provide unique challenges for financial LLMs evaluation. Novel datasets include (As shown in Table 2):

**FinTrade**. The FinTrade dataset is developed specifically for stock trading tasks, integrating historical stock prices, filings data, and news data for 10 stocks over a one-year period. It provides a robust foundation for evaluating LLMs in agent-based financial trading scenarios. The dataset is composed of three main components[6]: (1) **Stock Price Data**: Historical price data for 497 trading days, obtained via the yfinance API from Yahoo Finance, includes OHLCV (open, high, low, close, adjusted close price, and volume) metrics. Adjusted close prices are used to maintain consistency in the return series, minimizing the impact of corporate actions like dividends and stock splits. (2) **Filings Data**: Summary sections from Form 10-Q (quarterly reports) and Form 10-K (annual reports) are retrieved from the EDGAR database of the U.S. Securities and Exchange Commission (SEC). Over one year, each stock is linked to three quarterly reports and one annual report, providing crucial quarterly insights. (3) **News Data**: Daily news data, compiled from multiple publicly accessible datasets, provides short-term market perspectives, enabling the agent to account for market sentiment. The table below summarizes the data statistics.

**Regulations**. The Regulations dataset focuses on long-form question answering related to Over-the-Counter (OTC) derivatives and financial regulations within the European Union. Derived from the European Securities and Markets Authority's (ESMA) comprehensive document on Regulation (EU) No 648/2012 (EMIR), it maps QA pairs to relevant articles from EMIR and other directives. EMIR, implemented to enhance transparency and reduce risks in derivatives trading, governs OTC derivatives, central counterparties, and trade repositories. The dataset includes 254 QA pairs, meticulously curated with domain experts to ensure relevance and accuracy, addressing key regulatory issues such as reporting requirements, clearing thresholds, and obligations for financial and non-financial counterparties. The QAs are updated to reflect ongoing regulatory changes, providing a dynamic resource for testing LLMs' understanding of complex regulatory frameworks. This dataset serves as a critical tool for both regulatory compliance and academic research.

## 2.3 Tasks

Table 2 and Figure 1 shows all tasks, datasets, data statistics, and evaluation metrics covered by FinBen[7].

**Information extraction:** It spans seven datasets across six information extraction tasks. *1) Named entity recognition* extracts entities like LOCATION, ORGANIZATION, and PERSON from financial agreements and SEC filings, using the NER (Alvarado et al., 2015) and FINER-ORD (Shah et al., 2023b) datasets. *2) Relation extraction* identifies relationships such as "product/material produced" and "manufacturer" in financial news and earnings transcripts with the FINRED dataset (Sharma et al., 2022). *3) Causal classification* discerns whether sentences from financial news and SEC filings convey causality using the SC dataset (Mariko et al., 2020). *4) Causal detection* identifies cause and effect spans in financial texts with the CD dataset (Mariko et al., 2020). *5) Numeric labeling* tags numeric spans in financial documents using the FNXL dataset (Sharma et al., 2023), focusing on automating the assignment of labels from a large taxonomy to numeral spans in sentences. *6) Textual analogy parsing* involves identifying common attributes and comparative elements in textual analogies by extracting analogy frames, utilizing the FSRL dataset (Lamm et al., 2018), which maps analogous facts to semantic role representations and identifies the analogical relations between them. The evaluation of these tasks is focused on the F1 score (Goutte and Gaussier, 2005), Entity F1 score (Derczynski, 2016), and the Exact Match Accuracy (EM Accuracy) metric (Kim et al., 2023).

---

[6]Please see Appendix for more details

[7]For detailed instructions of each dataset, please see Appendix D

Table 2: The tasks, datasets, data statistics, and evaluation metrics included in FinBen. We use only test data for evaluation. Datasets marked with an asterisk (*) are newly constructed by us, comprising 10.32% of the total data. EM Accuracy means the exact match accuracy.

| Data | Task | Test | Evaluation | License |
|---|---|---|---|---|
| NER (Alvarado et al., 2015) | named entity recognition | 980 | Entity F1 | CC BY-SA 3.0 |
| FiNER-ORD (Shah et al., 2023b) | named entity recognition | 1,080 | Entity F1 | CC BY-NC 4.0 |
| FinRED (Sharma et al., 2022) | relation extraction | 1,068 | F1, Entity F1 | Public |
| SC (Mariko et al., 2020) | causal classification | 8,630 | F1, Entity F1 | CC BY 4.0 |
| CD (Mariko et al., 2020) | causal detection | 226 | F1, Entity F1 | CC BY 4.0 |
| FNXL (Sharma et al., 2023) | numeric labeling | 318 | F1, EM Accuracy | Public |
| FSRL (Lamm et al., 2018) | textual analogy parsing | 97 | F1, EM Accuracy | MIT License |
| FPB (Malo et al., 2014) | sentiment analysis | 970 | F1, Accuracy | CC BY-SA 3.0 |
| FiQA-SA (Maia et al., 2018) | sentiment analysis | 235 | F1 | Public |
| TSA (Cortis et al., 2017) | sentiment analysis | 561 | F1, Accuracy | CC BY-NC-SA 4.0 |
| Headlines (Sinha and Khandait, 2021) | news headline classification | 2,283 | Avg F1 | CC BY-SA 3.0 |
| FOMC (Shah et al., 2023a) | hawkish-dovish classification | 496 | F1, Accuracy | CC BY-NC 4.0 |
| FinArg-ACC (Sy et al., 2023) | argument unit classification | 969 | F1, Accuracy | CC BY-NC-SA 4.0 |
| FinArg-ARC (Sy et al., 2023) | argument relation classification | 496 | F1, Accuracy | CC BY-NC-SA 4.0 |
| MultiFin (Jørgensen et al., 2023) | multi-class classification | 690 | F1, Accuracy | Public |
| MA (Yang et al., 2020a) | deal completeness classification | 500 | F1, Accuracy | Public |
| MLESG (Chen et al., 2023a) | ESG Issue Identification | 300 | F1, Accuracy | CC BY-NC-ND |
| FinQA (Chen et al., 2021) | question answering | 1,147 | EM Accuracy | MIT License |
| TATQA (Zhu et al., 2021) | question answering | 1,668 | F1, EM Accuracy | MIT License |
| *Regulations | long-form question answering | 254 | ROUGE, BERTScore | Public |
| ConvFinQA (Chen et al., 2022b) | multi-turn question answering | 1,490 | EM Accuracy | MIT License |
| ECTSum (Mukherjee et al., 2022) | text summarization | 495 | ROUGE, BERTScore, BARTScore | Public |
| EDTSum (Xie et al., 2023b) | text summarization | 2,000 | ROUGE, BERTScore, BARTScore | Public |
| BigData22 (Soun et al., 2022) | stock movement prediction | 1,470 | Accuracy, MCC | Public |
| ACL18 (Xu and Cohen, 2018) | stock movement prediction | 3,720 | Accuracy, MCC | MIT License |
| CIKM18 (Wu et al., 2018) | stock movement prediction | 1,140 | Accuracy, MCC | Public |
| German (Hofmann, 1994) | credit scoring | 1,000 | F1, MCC | CC BY 4.0 |
| Australian (Quinlan, [n. d.]) | credit scoring | 690 | F1, MCC | CC BY 4.0 |
| LendingClub (Feng et al., 2023) | credit scoring | 2,690 | F1, MCC | CC0 1.0 |
| ccf (Feng et al., 2023) | fraud detection | 2,278 | F1, MCC | (DbCL) v1.0 |
| ccfraud (Feng et al., 2023) | fraud detection | 2,097 | F1, MCC | Public |
| polish (Feng et al., 2023) | financial distress identification | 1,736 | F1, MCC | CC BY 4.0 |
| taiwan (Feng et al., 2023) | financial distress identification | 1,364 | F1, MCC | CC BY 4.0 |
| ProtoSeguro (Feng et al., 2023) | claim analysis | 2,381 | F1, MCC | Public |
| travelinsurance (Feng et al., 2023) | claim analysis | 3,800 | F1, MCC | (ODbL) v1.0 |
| *FinTrade | stock trading | 3,384 | CR, SR, DV, AV, MD | MIT License |
| MultiFin-ES | multi-class classification | 2,066 | F1, Accuracy | MIT License |
| FNS-2023 | text summarization | 232 | ROUGE, BERTScore, BARTScore | Public |
| EFP | question answering | 37 | F1, Accuracy | Public |
| EFPA | question answering | 228 | F1, Accuracy | Public |
| TSA | sentiment analysis | 3,892 | F1, Accuracy | Public |
| FinanceES | sentiment analysis | 7,980 | F1, Accuracy | Public |

**Textual analysis:** This encompasses eight classification tasks for evaluating LLMs. *1) Sentiment analysis* focuses on extracting sentiment information (positive, negative, or neutral) from financial texts, using three datasets: the Financial Phrase Bank (FPB) (Malo et al., 2014), FiQA-SA (Maia et al., 2018), and TSA (Cortis et al., 2017). *2) News headline classification* analyzes additional information, like price movements in financial texts, using the Headlines dataset (Sinha and Khandait, 2021). *3) Hawkish-Dovish classification* aims to classify sentences from monetary policy texts as 'hawkish' or 'dovish' focusing on the nuanced language and economic implications of financial texts, using the FOMC (Shah et al., 2023a) dataset. *4) Argument unit classification* categorizes sentences as claims or premises using the FinArg AUC dataset (Sy et al., 2023). *5) Argument relation detection* identifies relationships (attack, support, or irrelevant) between social media posts using the FinArg ARC dataset (Sy et al., 2023). *6) Multi-class classification* targets categorizing a variety of financial texts, including analyst reports, news articles, and investor comments, utilizing the MultiFin dataset (Jørgensen et al., 2023). *7) Deal completeness classification* predicts if mergers and acquisitions events are "completed" or remain "rumors" based on news and tweets, employing the MA dataset (Yang et al., 2020a). *8) ESG issue identification* focuses on detecting Environmental, Social, and Governance (ESG) concerns in financial documents using the MLESG dataset (Chen et al., 2023a). For all datasets, evaluation utilizes the accuracy and F1 Score.

**Question answering.** It includes 4 datasets from three QA tasks, challenging LLMs to respond to financial queries. *1) Numerical QA* focuses on solving questions through multi-step numerical reasoning with financial reports and tables, utilizing the FinQA (Chen et al., 2021) and TATQA (Zhu et al., 2021) dataset. *2) Multi-turn QA* is an extension of QA with multi-turn questions and answers based on financial earnings reports and tables, using the ConvFinQA dataset (Chen et al., 2022b). F1

score (Derczynski, 2016) and the Exact Match Accuracy (EM Accuracy) metric (Kim et al., 2023) are used to evaluate these tasks. *3) Long-form QA* involves presenting models with complex, detailed questions that require extensive and nuanced answers, often incorporating legal interpretations and practical applications. In our evaluation, we utilize our newly proposed Regulations dataset, which focuses on intricate questions and answers related to financial regulations like EMIR. We assess the model responses using ROUGE (Lin, 2004) and BERTScore (Zhang et al., 2019).

**Text generation.** This task assesses the models' ability to produce coherent and informative text. Our focus is on *text summarization*, utilizing the ECTSUM (Mukherjee et al., 2022) dataset for summarizing earnings call transcripts. We also include EDTSUM, specifically designed for condensing financial news articles into concise summaries, constructed from original data in (Zhou et al., 2021). Evaluation employs ROUGE (Lin, 2004), BERTScore (Zhang et al., 2019), and BART Score (Yuan et al., 2021) to measure alignment, factual consistency, and information retention between machine-generated and expert summaries.

**Forecasting.** The forecasting task challenges models to predict future market and investor behaviors from emerging patterns. We focus on the *stock movement prediction* task, forecasting stock directions as either positive or negative, based on historical prices and tweets. Three datasets are included: BigData22 (Soun et al., 2022), ACL18 (Xu and Cohen, 2018) and CIKM18 (Wu et al., 2018).

**Risk management**. It challenges LLMs to accurately identify, extract, and analyze relevant risk-related information, interpret numerical data, and understand complex relationships. We include 4 tasks: *1) Credit scoring* classifies individuals as "good" or "bad" credit risks using historical customer data, employing datasets including: German (Hofmann, 1994), Australia (Quinlan, [n. d.]) and LendingClub (Feng et al., 2023). *2) Fraud detection* involve categorizes transactions as "fraudulent" or "non-fraudulent", using two datasets: ccf (Feng et al., 2023) and ccFraud (Feng et al., 2023). *3) Financial distress identification* aims to predict a company's bankruptcy risk, using the polish (Feng et al., 2023) and taiwan dataset (Feng et al., 2023). Note that the dataset name describes only the region of the company, and the content within the datasets is in English. *4) Claim analysis* anonymizes client data for privacy, labeling a "target" to indicate claim status, using two datasets: PortoSeguro (Feng et al., 2023) and travelinsurance (Feng et al., 2023). It is noticed that the dataset name such as German and taiwan, only indicates customer sources and all content is in English. F1 score and Matthews correlation coefficient (MCC) (Chicco and Jurman, 2020) are used for evaluating these tasks.

**Decision-making.** Strategic decision-making (Punt, 2017) evaluates the model's proficiency in synthesizing diverse information to formulate and implement trading strategies, a challenge even for experts. We innovatively introduce the SOTA financial LLM agent FinMem (Yu et al., 2023, 2024) to evaluate LLMs on the *stock trading* task. We construct the novel FinTrade dataset, containing 10 stocks, simulating real-world trading through historical prices, news, and sentiment analysis. Performance is measured by Cumulative Return (CR) (Ariel, 1987), Sharpe Ratio (SR) (Sharpe, 1998), Daily (DV) and Annualized volatility (AV) (Zhou et al., 2023), and Maximum Drawdown (MD) (Magdon-Ismail and Atiya, 2004), offering a comprehensive assessment of profitability, risk management, and decision-making prowess.

**Spanish.** Spanish financial datasets (Zhang et al., 2024) evaluate model performance in low-resource language settings. We include six datasets in our analysis: TSA-ES (Zhang et al., 2024) and FinanceES (Zhang et al., 2024), both designed for sentiment analysis in the Spanish financial domain, where model performance is measured using F1 score. For multi-class classification, we utilize the Spanish subset of the MultiFin dataset (Jørgensen et al., 2023), with F1 score as the primary metric. The EFP (Zhang et al., 2024) and EFPA (Zhang et al., 2024) datasets, focused on Spanish financial question-answering, are evaluated using F1 score to assess the accuracy of predicted answers. Finally, for summarization tasks, the FNS-2023 (Zhang et al., 2024) dataset, which consists of Spanish company reports, is evaluated using ROUGE scores to measure the quality of generated summaries.

## 3 Evaluation

We evaluate the zero-shot (from our evaluation) and few-shots (results from previous papers) performance of 21 representative general LLMs and financial LLMs on the FinBen benchmark, including: 1) ChatGPT: A LLM developed by OpenAI. 2) GPT-4 (OpenAI, 2023): The SOTA commercialized LLMs proposed by OpenAI. 3) Gemini Pro (Team et al., 2023): A multimodal LLM with 50T

parameters, released by Google. 4) LLaMA2-7/70B-chat (Touvron et al., 2023b): An open-sourced instruction-following LLM with 7B and 70B parameters developed by MetaAI. 5) LLaMA3-8B[8]: An open-sourced LLMs developed by MetaAI, using more training data than LLaMA2. 6) ChatGLM3-6B (Du et al., 2022): A conversational LLM with 6B parameters, jointly released by Zhipu AI and Tsinghua KEG. 7) Baichuan2-6B (Baichuan, 2023): An open-source LLM with 6B parameters, launched by Baichuan Intelligent Technology. 8) InternLM-7B (Team, 2023): An open-sourced 7B parameter base model tailored for practical scenarios, proposed by SenseTime. 9) Falcon-7B (Almazrouei et al., 2023): A 7B parameter causal decoder-only LLM model trained on 1500B tokens of RefinedWeb enhanced with curated corpora. 10) Mixtral 8×7B (Jiang et al., 2024): A LLM with the Sparse Mixture of Experts (SMoE) architecture. 11) Code LLaMA-7B (Roziere et al., 2023): An open-source LLM model for generating programming code, launched by Meta AI with 7B parameters. 12) FinGPT (Yang et al., 2023a): A 7B instruction finetuned financial LLM based on LLaMA 7B (Touvron et al., 2023a) with sentiment analysis tasks. 13) FinMA-7B (Xie et al., 2023b): A 7B instruction finetuned financial LLM based on LLaMA 7B with multiple NLP and forecasting tasks. 14) DISC-FinLLM (Chen et al., 2023b): An open-sourced financial LLM, fine-tuned from Baichuan-13B-Chat (Baichuan, 2023). 15) CFGPT (Li et al., 2023a): An open-source LLM, specifically designed for the financial sector and trained on Chinese financial datasets, which comprises 7B parameters. 16) Qwen2-7B/72B (qwe, 2024): Instruction-tuned LLMs developed by Alibaba Cloud with 7B and 72B parameters, optimized for financial and general NLP tasks. 17) Xuanyuan-6B/70B (Zhang et al., 2023c): Instruction-tuned LLMs designed for financial NLP tasks with 6B and 70B parameters. 18) LLaMA3.1-8B/70B (Dubey et al., 2024): LLaMA3 series models with 8B and 70B parameters, fine-tuned with enhanced data for a wide range of NLP tasks.

**Experimental Settings** We set the maximum generation tokens for LLMs to 1024 and the batch size to 20,000 for all experiments. These experiments are exclusively conducted on 16 NVIDIA A100 80G GPUs, taking approximately 600 hours to complete. Including the GPT-4 API costs, the total expenditure amounts to approximately $51,000.

# 4 Results

Table 3 and Table 4 shows the performance of 14 representative LLMs on all datasets in the FinBen. We also report results of non-LLM methods (traditional methods) in Appendix H.

## 4.1 Information Extraction and Textual Analysis Results

As shown in Table 3, for IE tasks, GPT-4 demonstrates superior performance in named entity recognition tasks, including NER, FINER-ORD, and FinRED. InternLM 7B achieves the best results in causal classification (SC). However, for more complex information extraction tasks, such as causal detection (CD) and numerical understanding (FNXL and FSRL), even GPT-4's performance is limited, with Gemini showing only slightly better results, still falling short of expectations. Additionally, while financial domain-specific LLMs developed by instruction tuning such as FinMA 7B exhibit improvements over general domain LLMs such as LLaMA2 7B-chat, they continue to struggle with both named entity recognition and complex extraction tasks. These findings highlight significant opportunities for advancement in financial causal detection and numerical understanding for LLMs.

Regarding TA tasks, instruction fine-tuned models like FinMA 7B exhibit the best performance in sentiment analysis tasks, including FPB, FiQA-SA, and Headlines. However, the generalization ability of FinMA 7B is limited due to the diversity of TA tasks in the financial domain. It performs even worse than general domain LLMs such as LLaMA2-7B-chat on other TA tasks, where GPT-4, Gemini, and LLaMA2 70B show superior results. This underscores the limitations of instruction fine-tuned models, which may be constrained by the parameter size and ability of their base models.

Models tailored for the Chinese language, such as CFGPT sft-7B-Full, which is fine-tuned on Chinese financial data, exhibit limited improvement on some datasets and even a decline in performance on others like MultiFin compared to its base model InternLM 7B. This trend suggests a language-based discrepancy, indicating that fine-tuning with Chinese data may adversely affect performance on English tasks. These findings underscore the complexities of cross-lingual adaptation in model training, highlighting the challenges in achieving consistent performance across different languages.

---

[8]https://llama.meta.com/llama3/

Table 3: The zero-shot and few-shot performance of different LLMs in FinBen. All results via our evaluations are the average of three runs. "-" represents the result that is currently unable to yield due to model size or availability, and "*" represents the result from the previous paper.

| Dataset | Metrics | Chat GPT | GPT 4 | Gemini | LLaMA2 7B-chat | LLaMA2 70B | LLaMA3 8B | FinMA 7B | FinGPT 7b-lora | InternLM 7B | Falcon 7B | Mixtral 7B | CFGPT sft-7B-Full |
|---|---|---|---|---|---|---|---|---|---|---|---|---|---|
| NER | EntityF1 | 0.77* | **0.83*** | 0.61 | 0.18 | 0.04 | 0.08 | 0.69 | 0.00 | 0.00 | 0.00 | 0.24 | 0.00 |
| FINER-ORD | EntityF1 | 0.28 | **0.77** | 0.14 | 0.02 | 0.07 | 0.00 | 0.00 | 0.00 | 0.00 | 0.00 | 0.05 | 0.00 |
| FinRED | F1 | 0.00 | **0.02** | 0.00 | 0.00 | 0.00 | 0.00 | 0.00 | 0.00 | 0.00 | 0.00 | 0.00 | 0.00 |
| SC | F1 | 0.80 | 0.81 | 0.74 | 0.85 | 0.61 | 0.69 | 0.19 | 0.00 | **0.88** | 0.67 | 0.83 | 0.15 |
| CD | F1 | 0.00 | 0.01 | **0.03** | 0.00 | 0.01 | 0.00 | 0.00 | 0.00 | 0.00 | 0.00 | 0.00 | 0.00 |
| FNXL | EntityF1 | 0.00 | 0.00 | 0.00 | 0.00 | 0.00 | 0.00 | 0.00 | 0.00 | 0.00 | 0.00 | 0.00 | 0.00 |
| FSRL | EntityF1 | 0.00 | 0.01 | **0.03** | 0.00 | 0.01 | 0.00 | 0.00 | 0.00 | 0.00 | 0.00 | 0.00 | 0.00 |
| FPB | F1 | 0.78* | 0.78* | 0.77 | **0.88** | 0.39 | 0.73 | 0.52 | **0.88** | 0.00 | 0.69 | 0.07 | 0.29 | 0.35* |
| | Acc | 0.78* | 0.76* | 0.77 | 0.41 | 0.72 | 0.52 | **0.88** | 0.00 | 0.69 | 0.05 | 0.37 | 0.26* |
| FiQA-SA | F1 | 0.60 | 0.80 | 0.81 | 0.76 | **0.83** | 0.70 | 0.79 | 0.00 | 0.81 | 0.77 | 0.16 | 0.42* |
| TSA | RMSE↓ | 0.53 | 0.50 | 0.37 | 0.71 | 0.57 | 0.25 | 0.80 | 0.00 | **0.29** | 0.50 | 0.50 | 1.05 |
| Headlines | AvgF1 | 0.77* | 0.86* | 0.78 | 0.72 | 0.63 | 0.60 | **0.97** | 0.60 | 0.60 | 0.45 | 0.60 | 0.61* |
| FOMC | F1 | 0.64 | **0.71** | 0.40 | 0.35 | 0.49 | 0.40 | 0.49 | 0.00 | 0.36 | 0.30 | 0.37 | 0.16* |
| | Acc | 0.6 | **0.69** | 0.60 | 0.49 | 0.47 | 0.41 | 0.46 | 0.00 | 0.35 | 0.30 | 0.35 | 0.21* |
| FinArg-ACC | MicroF1 | 0.50 | **0.60** | 0.31 | 0.46 | 0.58 | 0.51 | 0.27 | 0.00 | 0.39 | 0.23 | 0.39 | 0.05 |
| FinArg-ARC | MicroF1 | 0.39 | 0.40 | **0.60** | 0.27 | 0.36 | 0.28 | 0.08 | 0.00 | 0.33 | 0.32 | 0.57 | 0.05 |
| MultiFin | MicroF1 | 0.59 | **0.65** | 0.62 | 0.20 | 0.63 | 0.39 | 0.14 | 0.00 | 0.34 | 0.09 | 0.37 | 0.05 |
| MA | MicroF1 | 0.85 | 0.79 | 0.84 | 0.70 | **0.86** | 0.34 | 0.45 | 0.00 | 0.78 | 0.39 | 0.34 | 0.25 |
| MLESG | MicroF1 | 0.25 | **0.35** | 0.34 | 0.03 | 0.31 | 0.12 | 0.00 | 0.00 | 0.14 | 0.06 | 0.17 | 0.01 |
| FinQA | EmAcc | 0.58* | **0.63*** | 0.00 | 0.00 | 0.06 | 0.00 | 0.04 | 0.00 | 0.00 | 0.00 | 0.00 | 0.00 |
| TATQA | EmAcc | 0.00* | **0.13*** | 0.18 | 0.03 | 0.01 | 0.01 | 0.00 | 0.00 | 0.00 | 0.00 | 0.01 | 0.00 |
| Regulations | Rouge-1 | 0.12 | 0.11 | - | 0.24 | - | 0.10 | 0.12 | 0.01 | 0.04 | 0.03 | - | 0.14 |
| | BertScore | 0.64 | 0.62 | - | 0.65 | - | 0.60 | 0.59 | 0.40 | 0.57 | 0.14 | - | 0.57 |
| ConvFinQA | EmAcc | 0.60* | **0.76*** | 0.43 | 0.00 | 0.25 | 0.00 | 0.20 | 0.00 | 0.00 | 0.00 | 0.31 | 0.01 |
| EDTSUM | Rouge-1 | 0.17 | 0.20 | **0.39** | 0.17 | 0.25 | 0.14 | 0.13 | 0.00 | 0.13 | 0.15 | 0.12 | 0.01 |
| | BertScore | 0.66 | 0.67 | **0.72** | 0.62 | 0.68 | 0.60 | 0.38 | 0.52 | 0.48 | 0.57 | 0.61 | 0.51 |
| | BartScore | -3.64 | **-3.62** | -3.87 | -3.99 | -3.81 | -4.94 | -5.71 | -7.23 | -4.60 | -6.1 | -4.47 | -7.08 |
| ECTSUM | Rouge-1 | 0.00 | 0.00 | 0.00 | 0.00 | 0.00 | 0.00 | 0.00 | 0.00 | 0.00 | 0.00 | 0.00 | 0.00 |
| | BertScore | 0.00 | 0.00 | 0.00 | 0.00 | 0.00 | 0.00 | 0.00 | 0.00 | 0.00 | 0.00 | 0.00 | 0.00 |
| | BartScore | -5.18 | -5.18 | -4.93 | -5.18 | **-4.86** | -5.18 | -5.18 | -5.18 | -5.18 | --5.18 | -5.18 | -5.18 |
| BigData22 | Acc | 0.53 | 0.54 | **0.55** | 0.54 | 0.47 | 0.55 | 0.51 | 0.45 | 0.56 | 0.55 | 0.46 | 0.45 |
| | MCC | -0.025 | 0.03 | 0.04 | 0.05 | 0.00 | 0.02 | 0.02 | 0.00 | **0.08** | 0.00 | 0.02 | 0.03 |
| ACL18 | Acc | 0.50 | **0.52** | 0.52 | 0.51 | 0.51 | 0.52 | 0.51 | 0.49 | 0.51 | 0.51 | 0.49 | 0.48 |
| | MCC | 0.005 | 0.02 | **0.04** | 0.01 | 0.01 | 0.02 | 0.03 | 0.00 | 0.02 | 0.00 | 0.00 | -0.03 |
| CIKM18 | Acc | 0.55 | **0.57** | 0.54 | 0.55 | 0.49 | 0.57 | 0.50 | 0.42 | 0.57 | 0.47 | 0.42 | 0.41 |
| | MCC | 0.01 | 0.02 | 0.02 | -0.03 | -0.07 | 0.03 | **0.08** | 0.00 | -0.03 | -0.06 | -0.05 | -0.07 |
| German | F1 | 0.20 | 0.55 | 0.52 | **0.57** | 0.17 | 0.56 | 0.17 | 0.52 | 0.41 | 0.23 | 0.53 | 0.53 |
| | MCC | -0.10 | -0.02 | 0.00 | **0.03** | 0.00 | 0.05 | 0.00 | 0.00 | -0.30 | -0.07 | 0.00 | 0.00 |
| Australian | F1 | 0.41 | **0.74** | 0.26 | 0.26 | 0.41 | 0.26 | 0.41 | 0.38 | 0.34 | 0.26 | 0.26 | 0.29 |
| | MCC | 0.00 | **0.47** | 0.00 | 0.00 | 0.00 | 0.00 | 0.00 | 0.11 | 0.13 | 0.00 | 0.00 | -0.10 |
| LendingClub | F1 | 0.20 | 0.55 | 0.65 | **0.72** | 0.17 | 0.10 | 0.61 | 0.00 | 0.59 | 0.02 | 0.61 | 0.05 |
| | MCC | -0.10 | -0.02 | **0.19** | 0.00 | 0.00 | -0.15 | 0.00 | 0.00 | 0.15 | -0.01 | 0.08 | 0.01 |
| ccf | F1 | 0.20 | 0.55 | 0.96 | 0.00 | 0.17 | 0.01 | 0.00 | 1.00 | **1.00** | 0.10 | 0.00 | 0.00 |
| | MCC | -0.10 | -0.02 | -0.01 | **0.00** | 0.00 | 0.00 | 0.00 | 0.00 | 0.00 | 0.00 | 0.00 | 0.00 |
| ccfraud | F1 | 0.20 | 0.55 | **0.90** | 0.25 | 0.17 | 0.36 | 0.01 | 0.00 | 0.57 | 0.62 | 0.48 | 0.03 |
| | MCC | -0.10 | -0.02 | 0.00 | -0.16 | 0.00 | -0.03 | -0.06 | 0.00 | -0.13 | -0.02 | **0.16** | 0.01 |
| polish | F1 | 0.20 | 0.55 | 0.86 | 0.92 | 0.17 | 0.83 | 0.92 | 0.30 | 0.92 | 0.76 | **0.92** | 0.40 |
| | MCC | -0.10 | -0.02 | **0.14** | 0.00 | 0.00 | -0.06 | -0.01 | 0.00 | 0.07 | 0.05 | 0.00 | -0.02 |
| taiwan | F1 | 0.20 | 0.55 | **0.95** | 0.95 | 0.17 | 0.26 | 0.95 | 0.60 | 0.95 | 0.00 | 0.95 | 0.70 |
| | MCC | -0.10 | -0.02 | 0.00 | **-0.01** | 0.00 | -0.07 | 0.00 | 0.00 | -0.01 | 0.00 | 0.00 | 0.00 |
| portoseguro | F1 | 0.20 | 0.55 | 0.95 | 0.01 | 0.17 | 0.94 | 0.04 | **0.96** | 0.96 | 0.95 | 0.72 | 0.00 |
| | MCC | -0.10 | -0.02 | 0.00 | -0.05 | 0.00 | -0.01 | **0.01** | 0.00 | 0.00 | 0.00 | 0.01 | 0.00 |
| travelinsurance | F1 | 0.20 | 0.55 | 0.00 | 0.00 | 0.17 | 0.00 | 0.00 | **0.98** | 0.89 | 0.77 | 0.00 | 0.03 |
| | MCC | -0.10 | -0.02 | 0.00 | 0.00 | 0.00 | 0.00 | 0.00 | 0.00 | **0.12** | -0.03 | 0.00 | 0.01 |
| MultiFin-ES | ACC | 0.48 | **0.60** | 0.23 | 0.23 | 0.11 | 0.25 | 0.09 | 0.05 | 0.13 | 0.02 | 0.43 | 0.30 |
| | F1 | 0.47 | **0.60** | 0.14 | 0.11 | 0.12 | 0.27 | 0.12 | 0.07 | 0.17 | 0.03 | 0.42 | 0.27 |
| EFP | ACC | 0.30 | 0.27 | 0.27 | 0.27 | 0.27 | 0.35 | 0.27 | 0.27 | 0.27 | 0.24 | **0.41** | 0.27 |
| | F1 | **0.47** | 0.19 | 0.12 | 0.12 | 0.12 | 0.21 | 0.12 | 0.12 | 0.12 | 0.20 | 0.41 | 0.14 |
| EFPA | ACC | 0.31 | 0.34 | 0.25 | 0.26 | 0.20 | 0.35 | 0.25 | 0.26 | 0.25 | 0.23 | **0.38** | 0.32 |
| | F1 | 0.25 | 0.27 | 0.10 | 0.10 | 0.09 | 0.21 | 0.10 | 0.10 | 0.12 | 0.22 | **0.37** | 0.18 |
| FinanceES | ACC | 0.13 | 0.15 | 0.29 | 0.14 | 0.20 | 0.02 | 0.12 | 0.15 | 0.13 | 0.01 | **0.30** | 0.05 |
| | F1 | 0.08 | 0.09 | 0.16 | 0.13 | 0.23 | 0.03 | 0.16 | 0.18 | 0.20 | 0.02 | **0.30** | 0.05 |
| TSA | ACC | 0.21 | 0.47 | 0.40 | 0.07 | 0.03 | 0.04 | 0.02 | 0.06 | 0.001 | 0.02 | **0.53** | 0.07 |
| | F1 | 0.24 | 0.46 | 0.44 | 0.04 | 0.06 | 0.07 | 0.04 | 0.10 | 0.002 | 0.04 | **0.52** | 0.05 |
| FNS | Rouge-1 | 0.02 | 0.19 | **0.30** | 0.00 | 0.00 | 0.00 | 0.00 | 0.00 | 0.00 | 0.00 | 0.05 | 0.02 |
| | Rouge-2 | 0.04 | 0.06 | **0.06** | 0.00 | 0.00 | 0.00 | 0.00 | 0.00 | 0.00 | 0.00 | 0.05 | 0.00 |
| | Rouge-L | 0.12 | 0.13 | **0.16** | 0.00 | 0.00 | 0.00 | 0.00 | 0.00 | 0.00 | 0.00 | 0.05 | 0.02 |

## 4.2 Question Answering and Text Generation Results

In the QA tasks, closed-source commercial LLMs like GPT-4 and Gemini continue to lead across all datasets. While FinMA 7B shows improvement over its base models, it remains limited by model size and exhibits bottlenecks in numeric reasoning ability. For the regulations dataset, which is the first intersection dataset requiring both financial and legal knowledge, GPT-4 demonstrates its broad knowledge coverage effectively.

In the TG tasks, Gemini emerges as the frontrunner on the EDTSUM abstractive text summarization dataset, illustrating its prowess in generating coherent summaries. Nevertheless, all models face challenges with extractive summarization, which demands the generation of precise label sequences for

sentences. Among open-source LLMs, LLaMA2 70B stands out in text summarization. Conversely, CFGPT sft-7B-Full consistently shows a decrease in performance compared to its foundational model, InternLM 7B.

### 4.3 Forecasting and Risk Management Results

For forecasting, it is crucial to acknowledge that all LLMs fail to meet expected outcomes and lag behind traditional methodologies. This consistent observation with existing studies Xie et al. (2023b) underlines a notable deficiency in LLMs' capacity to tackle forecasting as effectively as traditional methods. Even the best-performing models, such as GPT-4 and Gemini, only perform slightly better than random guessing. This reveals significant potential for enhancement in LLMs, including industry leaders like GPT-4 and Gemini, particularly in forecasting tasks that demand complex reasoning abilities.

In RM tasks, such as credit scoring, fraud detection, and identifying financial distress, data often exhibit significant imbalances. Instances representing individuals with low credit scores, those prone to fraud, and companies at risk of financial distress constitute only a small percentage of the overall dataset. In such scenarios, LLMs with low instruction-following abilities (such as LLaMA2-7B-chat and LLaMA2-70B) tend to classify all cases into a single class, resulting in an MCC score of 0. These tasks, with tabular inputs and highly imbalanced distribution, pose a significant challenge for LLMs in the financial domain.

### 4.4 Decision Making Results

The comparative analysis of various LLMs on the complex task of stock trading, is presented in Table 4[9]. This task requires models to understand, summarize, and reason with multimodal financial data (texts and time series), leading to sophisticated trading decisions that necessitate a range of skills, from fundamental comprehension and summarization to reasoning and decision-making.

Among the evaluated LLMs, GPT-4 distinguishes itself by achieving the highest Sharpe Ratio (SR) over 1, indicating superior investment performance through optimal risk-return balance. It also records the minimal Maximum Drawdown (MDD), suggesting effective limitation of potential losses, thereby offering a more secure investment avenue compared to other models, including those using reinforcement learning methods like DQN, PPO, and A2C, which show significantly lower SR and higher MDD.

Tables 4 and 10 reinforce these findings, highlighting GPT-4's exceptional performance in this challenging domain. Additional results and analyses from these models in Table 5 contrast their performances with the traditional *Buy & Hold* strategy, which considerably lags behind.

Table 4: The average trading performance (95% Confidence Interval) comparison for different LLMs across 10 stocks. The results include large LLMs only ($\geq 70B$), as models with smaller contexts have difficulty understanding the instructions and producing a static strategy of holding.

| Model | CR (%)↑ | SR↑ | DV (%)↓ | AV (%)↓ | MD (%)↓ |
|---|---|---|---|---|---|
| Buy & Hold | -4.00 ± 22.39 | 0.02 ± 0.87 | 3.59 ± 1.34 | 56.43 ± 21.00 | 30.67 ± 17.48 |
| GPT-4 | **28.19 ± 25.27** | **1.51 ± 1.08** | 2.52 ± 1.30 | 39.88 ± 20.66 | **18.34 ± 9.77** |
| GPT-4o | -5.54 ± 19.12 | -0.19 ± 0.84 | 2.73 ± 1.30 | 43.62 ± 20.67 | 29.96 ± 18.89 |
| GPT3.5-Turbo | 4.48 ± 22.23 | 0.15 ± 0.82 | 2.84 ± 1.47 | 45.39 ± 23.35 | 28.83 ± 15.40 |
| llama2-70B | 4.02 ± 24.65 | 0.52 ± 1.48 | 2.18 ± 1.28 | 34.86 ± 20.38 | 25.55 ± 16.83 |
| llama3-70B | -2.57 ± 22.63 | -0.04 ± 1.19 | 2.71 ± 1.54 | 43.42 ± 24.65 | 29.31 ± 15.57 |
| gemini | 14.95 ± 28.04 | 1.03 ± 1.24 | **2.17 ± 1.39** | **34.67 ± 22.23** | 20.13 ± 11.36 |

In contrast, ChatGPT exhibits significantly lower performance metrics, indicating limitations in its financial decision-making capabilities. Gemini, on the other hand, secures the position of second-best performer, showcasing lower risk and volatility compared to GPT-4, yet maintaining commendable returns. When considering open-source models, LLaMA-70B, despite its lower volatility, yields the least profit among the LLMs, highlighting a trade-off between risk management and profitability.

---

[9]For detailed trading performance, please see Appendix F

Table 5: Traditional model performances on stock trading.

| Model | Cumulative Return | Sharpe Ratio | Standard Deviation | Annualized Volatility | Max Drawdown |
|-------|-------------------|--------------|--------------------|-----------------------|--------------|
| A2C | -4.2232 | -0.2586 | 2.7522 | 43.6898 | 30.5819 |
| PPO | -0.5586 | 0.0085 | 2.7531 | 43.7048 | 28.9496 |
| DQN | -2.9924 | -0.1656 | 2.7486 | 43.6319 | 31.78 |

For smaller models with parameters less than 70 billion, a marked inability to adhere to trading instructions consistently across transactions is noted. This is attributed to their limited comprehension, extraction capabilities, and constrained context windows. This limitation underscores the critical challenges smaller LLMs face in tasks requiring intricate financial reasoning and decision-making, thereby spotlighting the necessity for more advanced models to tackle decision making tasks effectively.

## 4.5 Spanish Results

Table 3 presents the performance of various models on six Spanish financial datasets, highlighting significant language disparities. ChatGPT, GPT-4 and Gemini show limited performance compared with English datasets. Mixtral 7B performs competitively, showing that the multilingual ability can improve language-specific tasks. Smaller models, particularly from the LLaMA family, struggle with domain complexities, reinforcing the importance of robust multilingual pretraining. While top models excel in sentiment analysis, all models underperform in summarization tasks on FNS, stressing the need for enhanced adaptation to specialized Spanish financial language.

## 5 Conclusion

In this work, we present FinBen, a comprehensive benchmark specifically designed to evaluate LLMs in the financial domain. FinBen includes 42 diverse datasets spanning 24 tasks, meticulously organized to assess LLMs across eight critical aspects: information extraction, textual analysis, question answering, text generation, risk management, forecasting, decision-making, and Spanish. This breadth of coverage sets FinBen apart from existing financial benchmarks, enabling a more robust and nuanced evaluation of LLM capabilities. Our evaluation of 21 LLMs, including GPT-4, ChatGPT, and Gemini, reveals their key advantages and limitations, highlighting directions for future work. Looking ahead, FinBen continuously evolves into an open FinLLM leaderboard (Lin et al., 2024). We will incorporat additional languages and multimodal financial tasks (Yanglet and Deng, 2024) and expand the range of financial tasks to further enhance its applicability and impact.

**Openness**: Our FinBen project follows the model openness framework (White et al., 2024) by providing a comprehensive set of financial datasets and evalution codes under OSI-approved licenses.

**Limitations**: We acknowledge several limitations that could impact FinBen's effectiveness and applicability. The restricted size of available datasets may affect the models' financial understanding and generalization across various contexts. Computational constraints limited our evaluation to the LLaMA 70B model, potentially overlooking the capabilities of larger models. Additionally, the tasks are based on American market data and English texts, which may limit the benchmark's applicability to global financial markets. Responsible usage and safeguards are essential to prevent potential misuse, such as financial misinformation or unethical market influence[10].

**Ethical Statement**: The authors take full responsibility for any potential legal issues arising from FinBen's development and dissemination. All data used are publicly available, non-personal, and shared under the MIT license, adhering to privacy and ethical guidelines. This manuscript and associated materials are for academic and educational use only and do not provide financial, legal, or investment advice. The authors disclaim any liability for losses or damages from using the material, and users agree to seek professional consultation and indemnify the authors against any claims arising from its use[11].

---

[10]For a detailed limitation concerning this work, please see Appendix.

[11]For a detailed ethical and legal statement concerning this work, please see Appendix.

## Acknowledgements

The authors acknowledge UFIT Research Computing, NVAITC, and HPG for providing computational resources and support that have contributed to the research results reported in this publication. URL: http://www.rc.ufl.edu. This work is supported by the project JPNP20006 from New Energy and Industrial Technology Development Organization (NEDO). This work has also been partially supported by project MIS 5154714 of the National Recovery and Resilience Plan Greece 2.0 funded by the European Union under the Next Generation EU Program. Additionally, we gratefully acknowledge FINOS (Fintech Open Source Foundation) for supporting the Open Financial LLM Leaderboard initiative. Xiao-Yang Liu acknowledges the support from NSF IUCRC CRAFT center research grant (CRAFT Grant 22017) for this research. The opinions expressed in this publication do not necessarily represent the views of NSF IUCRC CRAFT. Haoqiang Kang and Xiao-Yang Liu also acknowledge the support from Columbia's SIRS and STAR Program, The Tang Family Fund for Research Innovations in FinTech, Engineering, and Business Operations.

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

# A    Contributions

**Science Leadership**: Qianqian Xie, Min Peng, Sophia Ananiadou, Alejandro Lopez-Lira, Hao Wang, Yanzhao Lai, Benyou Wang, Xiao-Yang Liu, Gang Hu, Jiajia Huang, Jimin Huang.

**Contributors**: Mengxi Xiao, Dong Li, Weiguang Han, Zhengyu Chen, Ruoyu Xiang, Xiao Zhang, Yueru He, Yongfu Dai, Duanyu Feng, Yijing Xu, Haoqiang Kang, Ziyan Kuang, Chenhan Yuan, Kailai Yang, Zheheng Luo, Tianlin Zhang, Zhiwei Liu, Guojun Xiong, Zhiyang Deng, Yuechen Jiang, Zhiyuan Yao, Haohang Li, Yangyang Yu

# B    Fintrade Dataset

Table 6: Summary of FinTrade dataset statistics.

| Ticker | Number of News | Number of 10-K/10-Q Files | Numerical Price Data |
|---|---|---|---|
| TSLA | 3,233 | 8 | 497 |
| NFLX | 965 | 8 | 497 |
| AMZN | 1,675 | 8 | 497 |
| MSFT | 1,362 | 8 | 497 |
| AAPL | 2,082 | 8 | 497 |
| GOOG | 1,144 | 7 | 497 |
| DIS | 1,445 | 9 | 497 |
| GM | 2,252 | 9 | 497 |
| NIO | 957 | 0 | 497 |
| COIN | 1,022 | 0 | 497 |

# C    Other LLMs Performance

Table 7 presents other LLMs' performance in the FinBen.

Table 7: The zero-shot and few-shots performance of other LLMs on the FinBen.

| Dataset | Metrics | Baichuan 7B | CodeLLaMA 7B | DISC-FinLLM | ChatGLM3 6B | Qwen2 7B | Xuanyuan 6B | Qwen2 72B | Xuanyuan 70B | LLaMA3.1 8B | LLaMA3.1 70B |
|---|---|---|---|---|---|---|---|---|---|---|---|
| NER | EntityF1 | 0.00 | 0.07 | 0.12 | **0.25** | 0.07 | 0.06 | 0.02 | 0.08 | 0.14 | 0.05 |
| FINER-ORD | EntityF1 | 0.00 | 0.00 | 0.00 | 0.02 | 0.02 | 0.02 | 0.02 | **0.33** | 0.12 | 0.18 |
| FinRED | F1 | 0.00 | 0.00 | 0.00 | 0.00 | 0.00 | 0.00 | 0.00 | **0.01** | 0.00 | 0.00 |
| SC | F1 | 0.74 | 0.85 | 0.00 | 0.81 | 0.60 | 0.70 | 0.82 | 0.23 | 0.83 | **0.87** |
| CD | F1 | **0.00** | 0.00 | 0.00 | 0.00 | 0.00 | 0.00 | 0.01 | 0.00 | 0.00 | 0.00 |
| FNXL | EntityF1 | **0.00** | 0.00 | 0.00 | 0.00 | 0.00 | 0.00 | 0.00 | 0.00 | 0.00 | 0.00 |
| FSRL | EntityF1 | **0.00** | 0.00 | 0.00 | 0.00 | 0.00 | 0.00 | 0.01 | 0.00 | 0.00 | 0.00 |
| FPB | F1 | 0.17 | 0.34 | 0.29 | 0.74 | 0.52 | 0.74 | 0.75 | 0.52 | 0.76 | **0.79** |
|  | Acc | 0.23 | 0.39 | 0.26 | 0.74 | 0.52 | 0.75 | 0.74 | 0.55 | 0.75 | **0.79** |
| FiQA-SA | F1 | 0.32 | 0.66 | 0.32 | 0.56 | 0.57 | 0.56 | 0.63 | **0.82** | 0.75 | 0.74 |
| TSA | RMSE↓ | 0.44 | 0.43 | 0.32 | 0.35 | 0.43 | 0.33 | 0.30 | 0.54 | **0.17** | 0.42 |
| Headlines | AvgF1 | 0.60 | 0.60 | 0.60 | 0.66 | 0.60 | 0.65 | 0.60 | **0.73** | 0.60 | 0.60 |
| FOMC | F1 | 0.17 | 0.14 | 0.19 | 0.47 | 0.63 | 0.45 | **0.65** | 0.60 | 0.48 | 0.64 |
|  | Acc | 0.25 | 0.27 | 0.28 | 0.46 | 0.64 | 0.51 | 0.66 | 0.61 | 0.56 | **0.67** |
| FinArg-ACC | MicroF1 | 0.36 | 0.28 | 0.29 | 0.25 | 0.43 | 0.47 | 0.57 | 0.58 | 0.53 | **0.65** |
| FinArg-ARC | MicroF1 | 0.27 | 0.25 | 0.29 | 0.50 | 0.53 | 0.60 | 0.63 | **0.67** | 0.55 | 0.55 |
| MultiFin | MicroF1 | 0.12 | 0.21 | 0.29 | 0.47 | 0.39 | 0.54 | 0.55 | 0.63 | 0.62 | **0.69** |
| M&A | MicroF1 | 0.33 | 0.54 | 0.29 | 0.79 | 0.83 | 0.84 | 0.84 | 0.79 | **0.85** | 0.84 |
| MLESG | MicroF1 | 0.04 | 0.10 | 0.29 | 0.16 | 0.34 | 0.24 | 0.43 | 0.26 | 0.31 | **0.44** |
| FinQA | EmAcc | **0.00** | 0.00 | 0.00 | 0.00 | 0.00 | 0.00 | 0.00 | 0.00 | 0.00 | 0.00 |
| TATQA | EmAcc | 0.00 | 0.00 | 0.00 | 0.07 | 0.00 | 0.11 | 0.00 | 0.02 | 0.04 | **0.44** |
| Regulations | Rouge-1 | 0.13 | - | - | 0.26 | **0.31** | 0.24 | 0.31 | 0.30 | 0.27 | 0.10 |
|  | BertScore | 0.60 | - | - | 0.65 | 0.68 | 0.64 | **0.69** | 0.67 | 0.65 | 0.61 |
| ConvFinQA | EmAcc | 0.00 | 0.00 | 0.00 | 0.00 | 0.00 | 0.00 | **0.01** | 0.00 | 0.00 | 0.00 |
| EDTSUM | Rouge-1 | 0.02 | 0.10 | 0.22 | 0.13 | 0.22 | 0.24 | 0.22 | **0.25** | 0.20 | 0.18 |
|  | BertScore | 0.47 | 0.67 | 0.61 | 0.47 | 0.67 | 0.66 | 0.67 | **0.68** | 0.64 | 0.63 |
| ECTSUM | Rouge-1 | **0.00** | 0.00 | 0.00 | 0.00 | 0.00 | 0.00 | 0.00 | 0.00 | 0.00 | 0.00 |
|  | BertScore | **0.00** | 0.00 | 0.00 | 0.00 | 0.00 | 0.00 | 0.00 | 0.00 | 0.00 | 0.00 |
| BigData22 | Acc | 0.53 | 0.52 | 0.44 | 0.47 | 0.55 | 0.50 | **0.56** | 0.56 | 0.54 | 0.45 |
|  | MCC | -0.01 | -0.01 | -0.05 | 0.00 | 0.01 | 0.00 | 0.05 | **0.06** | 0.03 | -0.00 |
| ACL18 | Acc | 0.50 | 0.51 | 0.50 | 0.50 | 0.50 | 0.51 | 0.50 | 0.49 | **0.52** | 0.49 |
|  | MCC | -0.01 | 0.00 | 0.02 | 0.02 | -0.02 | 0.01 | -0.01 | -0.02 | **0.05** | 0.03 |
| CIKM18 | Acc | 0.48 | 0.51 | 0.44 | 0.42 | 0.52 | 0.50 | 0.55 | 0.50 | **0.57** | 0.44 |
|  | MCC | **0.02** | 0.02 | -0.03 | 0.02 | -0.03 | -0.03 | -0.01 | 0.00 | -0.01 | 0.02 |

## D  Instructions

For detail instruction of each dataset, please see Table 8 and Table 9.

## E  Related Work

### E.1  Financial Large Language Models

Recent years have seen a significant surge in research on finance-specific LLMs, expanding on the groundwork laid by general-purpose language models (Lee et al., 2024; Liu et al., 2023b; Xie et al., 2023a; Zhang et al., 2024; Dai et al., 2024; Xie et al., 2024). Financial pre-trained language models (FinPLMs) like FinBERT (Araci, 2019; Yang et al., 2020b; Liu et al., 2020), derived from BERT, and FLANG (Shah et al., 2022), based on ELECTRA, have been developed using domain-specific data for enhanced performance in tasks like sentiment analysis and stock prediction. The open-source release of Meta AI's LLaMA (Touvron et al., 2023a,b) has fueled further innovation in Financial LLMs (FinLLMs), with models like FinMA (Xie et al., 2023b), InvestLM (Yang et al., 2023b), and FinGPT (Liu et al., 2023a, 2024a; Yang et al., 2023a) leveraging advanced tuning strategies (Zhang et al., 2023a) for financial applications. BloombergGPT (Wu et al., 2023) stands out as a BLOOM-based, closed-source model tailored for the financial industry. Additionally, the Chinese financial sector has seen the emergence of models like XuanYuan 2.0 (Zhang et al., 2023c), integrating broad and specialized knowledge, FinBART (Hongyuan et al., 2023) for financial communication, and CFGPT (Li et al., 2023a), which includes a comprehensive dataset for targeted pre-training and fine-tuning.

### E.2  Financial Evaluation Benchmarks

Financial evaluation benchmarks, such as the pioneering FLUE (Shah et al., 2022), have been introduced to measure model performance in the financial sector, covering five key NLP tasks: financial sentiment analysis (Shah et al., 2022), news headline classification (Sinha and Khandait, 2021), named entity recognition (NER) (Salinas Alvarado et al., 2015), structure boundary detection and question answering (QA) (Chen et al., 2022a). Building upon FLUE, FLARE (Xie et al., 2023b) added the evaluation of time-series processing capabilities, i.e., forecasting stock price movements. In addition, in Chinese financial benchmarks, there are more recently released Chinese datasets like CFBenchmark (Lei et al., 2023), DISC-FINSFT (Chen et al., 2023b), and CGCE (Zhang et al., 2023b). However, these benchmarks have a limited scope and have not yet addressed more complex financial NLP tasks such as event detection (Zhou et al., 2021), and realistic financial tasks, despite the fact that there were previous efforts on stock trading (Liu et al., 2022; Han et al., 2023a,b).

## F  Trading Accumulative Returns

Table 10 and the figures below show detailed trading performance.

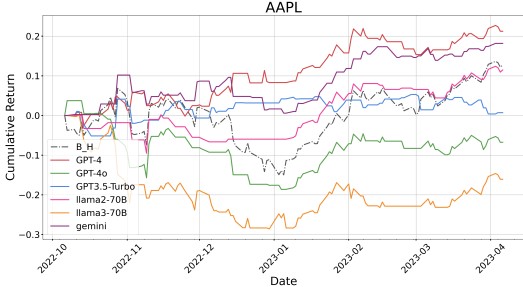

Figure 2: Accumulative Returns of LLM Trading Strategies on AAPL

Table 8: Quantification task datasets prompt overview.

| Data | Prompt |
|------|--------|
| FPB | "Analyze the sentiment of this statement extracted from a financial news article. Provide your answer as either negative, positive or neutral. For instance, 'The company's stocks plummeted following the scandal.' would be classified as negative." |
| FiQA-SA | "What is the sentiment of the following financial {category}: Positive, Negative, or Neutral?" |
| Headlines | "Consider whether the headline mentions the price of gold. Is there a Price or Not in the gold commodity market indicated in the news headline? Please answer Yes or No." |
| NER | "In the sentences extracted from financial agreements in U.S. SEC filings, identify the named entities that represent a person ('PER'), an organization ('ORG'), or a location ('LOC'). The required answer format is: 'entity name, entity type'. For instance, in 'Elon Musk, CEO of SpaceX, announced the launch from Cape Canaveral.', the entities would be: 'Elon Musk, PER; SpaceX, ORG; Cape Canaveral, LOC'" |
| FiNER-ORD | "In the list of tokens, identify {tid}each accordingly. If the entity spans multiple tokens, use the prefix B-PER, B-LOC, or B-ORG for the first token, and I-PER, I-LOC, or I-ORG for the subsequent tokens of that entity. The beginning of each separate entity should always be labeled with a B-PER, B-LOC, or B-ORG prefix. If the token does not fit into any of the three named categories, or is not a named entity, label it as 'O'." |
| FinQA | "Given the financial data and expert analysis, please answer this question:" |
| Regulations | "Please answer following questions." |
| ConvFinQA | "In the context of this series of interconnected finance-related queries and the additional information provided by the pretext, table data, and post text from a company's financial filings, please provide a response to the final question. This may require extracting information from the context and performing mathematical calculations. Please take into account the information provided in the preceding questions and their answers when formulating your response:" |
| BigData22 | " Contemplate the data and tweets to guess whether the closing price of {tid} will surge or decline at {point}. Please declare with either Rise or Fall." |
| ACL18 | "Scrutinize the data and tweets to envisage if the closing price of {tid}will swell or contract at {point}. Respond with either Rise or Fall." |
| CIKM18 | "Reflect on the provided data and tweets to anticipate if the closing price of {tid}is going to increase or decrease at {point}. Respond with either Rise or Fall." |
| ECTSum | "Given the following article, please produce a list of 0 and 1, each separated by ' ' to indicate which sentences should be included in the final summary. The article's sentences have been split by ' '. Please mark each sentence with 1 if it should be included in the summary and 0 if it should not." |
| EDTSum | "You are given a text that consists of multiple sentences. Your task is to perform abstractive summarization on this text. Use your understanding of the content to express the main ideas and crucial details in a shorter, coherent, and natural sounding text." |
| German | "Assess the creditworthiness of a customer using the following table attributes for financial status. Respond with either 'good' or 'bad'. And the table attributes including 13 categorical attributes and 7 numerical attributes are as follows:" |
| Australian | "Assess the creditworthiness of a customer using the following table attributes for financial status. Respond with either 'good' or 'bad'. And the table attributes including 13 categorical attributes and 7 numerical attributes and values have been changed to meaningless symbols to protect confidentiality of the data. :" |
| FOMC | "Examine the excerpt from a central bank's release below. Classify it as HAWKISH if it advocates for a tightening of monetary policy, DOVISH if it suggests an easing of monetary policy, or NEUTRAL if the stance is unbiased. Your response should return only HAWKISH, DOVISH, or NEUTRAL." |
| TSA | "Given the following financial text, return a sentiment score for Ashtead as a floating-point number ranging from -1 (indicating a very negative or bearish sentiment) to 1 (indicating a very positive or bullish sentiment), with 0 designating neutral sentiment. Return only the numerical score first, follow it with a brief reasoning behind your score." |
| FinArg - ACC | "Analyze sentences from earnings conference calls and identify their argumentative function. Each sentence is either a premise, offering evidence or reasoning, or a claim, asserting a conclusion or viewpoint. Return only premise or claim." |
| FinArg - ARC | "In this task, you are given a pair of sentences. Your objective is to ascertain the type of argumentative relation between these two sentences. The relation could either be 'NoRelation', indicating no discernible relation between the sentences, 'Support', indicating that the first sentence supports the second, or 'Attack', indicating that the first sentence disputes or contradicts the second. Return only one of the three classifications: 'norelation', 'support', or 'attack'." |
| MultiFin | "In this task, you're working with English headlines from the MULTIFIN dataset. This dataset is made up of real-world article headlines from a large accounting firm's websites. Your objective is to categorize each headline according to its primary topic. The potential categories are {category}. Your response should only include the category that best fits the headline." |
| MA | "In this task, you will be given Mergers and Acquisitions news articles or tweets. Your task is to classify each article or tweet based on whether the mentioned deal was completed or remained a rumour. Your response should be a single word - either 'complete' or 'rumour' - representing the outcome of the deal mentioned in the provided text." |
| MLESG | "You're given English news articles related to Environmental, Social, and Corporate Governance (ESG) issues. Your task is to classify each article based on the ESG issue it pertains to, according to the MSCI ESG rating guidelines. The ESG issues include {category}. Your output should be the most relevant ESG issue label, followed by a brief rationale based on the article content." |

Table 9: The example prompts of remaining tasks. FiQA-SA has two types of text, including news headlines and tweets. We will fill the detailed text type into {category} for each data sample. For stock movement prediction data such as BigData22, we will fill {tid} and {point} with the detailed stock name and time from each data sample. For Spanish tasks, please refer to (Zhang et al., 2024).

| Data | Prompt |
|---|---|
| FinRED | "Given the following sentence, identify the head, tail, and relation of each triplet present in the sentence. The relations you should be looking for are {category}. If a relation exists between two entities, provide your answer in the format {category}. If there are multiple triplets in a sentence, provide each one on a new line." |
| SC | "In this task, you are provided with sentences extracted from financial news and SEC data. Your goal is to classify each sentence into either 'causal' or 'noise' based on whether or not it indicates a causal relationship between financial events. Please return only the category 'causal' or 'noise'." |
| CD | "Your job in this task is to perform sequence labeling on a provided text section, marking the chunks that represent the cause of an event and the effects that result from it. For each token in the text, assign a label to indicate its role in representing cause or effect. The labels you should use are 'B-CAUSE', 'I-CAUSE', 'B-EFFECT', 'I-EFFECT', and 'O'. A 'B-' prefix is used to denote the beginning of a cause or effect sequence, while an 'I-' prefix is used for continuation of a cause or effect sequence. If a token is not part of either a cause or effect sequence, label it as 'O'. Provide your answer as a sequence of 'token:label' pairs, with each pair on a new line." |
| TATQA | "Please answer the given financial question based on the context. Context: {context}Question: What is the amount of total sales in 2019?" |
| FNXL | "In the task of Financial Numeric Extreme Labelling (FNXL), your job is to identify and label the semantic role of each token in a sentence. The labels can include {category}" |
| FSRL | "In the task of Textual Analogy Parsing (TAP), your job is to identify and label the semantic role of each token in a sentence. The labels can include {category}." |
| LendingClub | "Assess the client's loan status based on the following loan records from Lending Club. Respond with only 'good' or 'bad', and do not provide any additional information. For instance, 'The client has a stable income, no previous debts, and owns a property.' should be classified as 'good'." |
| ccf | "Detect the credit card fraud using the following financial table attributes. Respond with only 'yes' or 'no', and do not provide any additional information. Therein, the data contains 28 numerical input variables V1, V2, ..., and V28 which are the result of a PCA transformation and 1 input variable Amount which has not been transformed with PCA. The feature 'Amount' is the transaction Amount, this feature can be used for example-dependant cost-sensitive learning. For instance, 'The client has attributes:{category}'" |
| ccfraud | "Detect the credit card fraud with the following financial profile. Respond with only 'good' or 'bad', and do not provide any additional information. For instance, 'The client is a female, the state number is 25, the number of cards is 1, the credit balance is 7000, the number of transactions is 16, the number of international transactions is 0, the credit limit is 6.' should be classified as 'good'." |
| polish | "Predict whether the company will face bankruptcy based on the financial profile attributes provided in the following text. Respond with only 'no' or 'yes', and do not provide any additional information." |
| taiwan | "Predict whether the company will face bankruptcy based on the financial profile attributes provided in the following text. Respond with only 'no' or 'yes', and do not provide any additional information." |
| Porto-Seguro | "Identify whether or not to files a claim for the auto insurance policy holder using the following table attributes about individual financial profile. Respond with only 'yes' or 'no', and do not provide any additional information. And the table attributes that belong to similar groupings are tagged as such in the feature names (e.g., ind, reg, car, calc). In addition, feature names include the postfix bin to indicate binary features and cat to indicate categorical features. Features without these designations are either continuous or ordinal. Values of -1 indicate that the feature was missing from the observation." |
| travelinsurace | "Identify the claim status of insurance companies using the following table attributes for travel insurance status. Respond with only 'yes' or 'no', and do not provide any additional information. And the table attributes including 5 categorical attributes and 4 numerical attributes are as follows:{category}" |
| FinTrade | "Given the information, can you make an investment decision? Just summarize the reason of the decision. please consider only the available short-term information, the mid-term information, the long-term information, the reflection-term information. please consider the momentum of the historical stock price. When cumulative return is positive or zero, you are a risk-seeking investor. But when cumulative return is negative, you are a risk-averse investor. please consider how much share of the stock the investor holds now. You should provide exactly one of the following investment decisions: buy or sell. When it is really hard to make a 'buy'-or-'sell' decision, you could go with 'hold' option. You also need to provide the id of the information to support your decision. {investment_info} {gr:complete_json_suffix_v2} Your output should strictly conforms the following json format without any additional contents: {"investment_decision" : string, "summary_reason": string, "short_memory_index": number, "middle_memory_index": number, "long_memory_index": number, "reflection_memory_index": number}" |

# G FinLLM challenge

Based on our proposed FinBen, we organized the FinLLM Share Task during the FinNLP-AgentScen Workshop at IJCAI 2024[12], known as the FinLLM Challenge. This challenge not only tests the abilities of LLMs but also promotes ongoing research into their application within the financial sector, highlighting FinBen's critical contribution to the advancement of financial analytics.

---

[12]https://sites.google.com/nlg.csie.ntu.edu.tw/finnlp-agentscen/shared-task-finllm?authuser=0

Table 10: The overall trading performance comparison for different LLMs across various stocks. The results include large LLMs only ($\geq 70B$), as models with smaller contexts have difficulty understanding the instructions and producing a static strategy of holding.

| Ticker | Model | CR (%) | SR | DV (%) | AV (%) | MD (%) |
|--------|-------|--------|-----|--------|--------|--------|
| TSLA | Buy & Hold | -25.2137 | -0.7203 | 4.4099 | 70.0043 | 57.6765 |
| | GPT-4 | **68.3089** | **2.8899** | 2.9780 | **47.2739** | **10.7996** |
| | GPT-4o | -0.8789 | -0.0321 | 3.4531 | 54.8156 | 44.6842 |
| | GPT3.5-Turbo | 25.2137 | 0.7203 | 4.4099 | 70.0043 | 51.3186 |
| | llama2-70B | -31.4144 | -1.0412 | 3.8014 | 60.3450 | 48.6173 |
| | llama3-70B | -16.4424 | -0.4847 | 4.2743 | 67.8519 | 55.5486 |
| | gemini | -0.3790 | -0.0148 | 3.2271 | 51.2280 | 35.6707 |
| NFLX | Buy & Hold | 34.6251 | 1.3696 | 3.1852 | 50.5634 | 20.9263 |
| | GPT-4 | **36.4485** | **2.0088** | 2.2860 | 36.2894 | **15.8495** |
| | GPT-4o | 5.5829 | 0.2592 | 2.7132 | 43.0702 | 17.4715 |
| | GPT3.5-Turbo | 7.9337 | 0.4610 | 2.1680 | 34.4160 | 17.9578 |
| | llama2-70B | 33.8460 | 1.4741 | 2.8928 | 45.9216 | 20.3910 |
| | llama3-70B | 21.7374 | 0.9513 | 2.8788 | 45.6989 | 21.3478 |
| | gemini | 11.6298 | 1.0073 | **1.4546** | **23.0906** | 16.5106 |
| AMZN | Buy & Hold | -16.4428 | -0.7448 | 2.7812 | 44.1508 | 33.8847 |
| | GPT-4 | 10.5539 | 0.4923 | 2.7012 | 42.8802 | 22.9294 |
| | GPT-4o | 11.3626 | 0.7334 | 1.9520 | 30.9864 | 19.5964 |
| | GPT3.5-Turbo | **19.9636** | 0.9611 | 2.6171 | 41.5454 | 19.2191 |
| | llama2-70B | 8.3595 | **1.9715** | **0.5342** | **8.4804** | **0.0000** |
| | llama3-70B | 11.1479 | 0.5405 | 2.5986 | 41.2509 | 28.2174 |
| | gemini | -2.3838 | -0.5321 | 0.5645 | 8.9605 | 6.4291 |
| MSFT | Buy & Hold | 17.2161 | 0.9710 | 2.2339 | 35.4623 | 15.0097 |
| | GPT-4 | 25.7826 | **1.5818** | 2.0535 | 32.5989 | **14.9889** |
| | GPT-4o | -5.3731 | -0.5209 | **1.2997** | **20.6317** | 18.8223 |
| | GPT3.5-Turbo | 20.4179 | 1.3600 | 1.8915 | 30.0259 | 20.3212 |
| | llama2-70B | **27.7664** | 1.5708 | 2.2270 | 35.3524 | 15.0097 |
| | llama3-70B | 21.1983 | 1.2628 | 2.1149 | 33.5724 | 15.0097 |
| | gemini | 21.5081 | 1.3701 | 1.9777 | 31.3957 | 17.5051 |
| AAPL | Buy & Hold | 12.7371 | 0.7759 | 2.0682 | 32.8323 | 20.6590 |
| | GPT-4 | **21.2335** | **1.9274** | 1.3879 | 22.0328 | 6.4237 |
| | GPT-4o | -6.7540 | -0.5693 | 1.4948 | 23.7285 | 20.7600 |
| | GPT3.5-Turbo | 0.7110 | 0.0758 | **1.1817** | **18.7581** | **6.0818** |
| | llama2-70B | 11.4856 | 1.1550 | 1.2529 | 19.8885 | 9.2776 |
| | llama3-70B | -16.0835 | -1.1985 | 1.6907 | 26.8394 | 25.9520 |
| | gemini | 18.1718 | 1.7214 | 1.3300 | 21.1134 | 9.6467 |
| GOOG | Buy & Hold | 6.3107 | 0.3081 | 2.5806 | 40.9660 | 21.1907 |
| | GPT-4 | 13.2811 | 0.9667 | 1.7308 | 27.4762 | 12.2209 |
| | GPT-4o | 16.5072 | 1.0654 | 1.9520 | 30.9872 | **11.8863** |
| | GPT3.5-Turbo | 0.9990 | 0.0614 | 2.0490 | 32.5265 | 20.9316 |
| | llama2-70B | 17.0030 | **1.1057** | 1.9374 | 30.7546 | 13.2088 |
| | llama3-70B | 17.5630 | 0.8872 | 2.4942 | 39.5934 | 19.2783 |
| | gemini | **38.7956** | 3.0341 | **1.6110** | **25.5732** | 13.7311 |
| DIS | Buy & Hold | -0.0700 | -0.0037 | 2.3667 | 37.5695 | 22.7722 |
| | GPT-4 | **31.3383** | **2.3931** | 1.6498 | 26.1904 | 12.3417 |
| | GPT-4o | -20.2500 | -1.3737 | 1.8573 | 29.4830 | 27.0246 |
| | GPT3.5-Turbo | -7.1533 | -0.5109 | 1.7641 | 28.0048 | 20.4278 |
| | llama2-70B | -3.8257 | -1.4323 | **0.3365** | **5.3420** | **4.1451** |
| | llama3-70B | -25.5829 | -1.5579 | 2.0690 | 32.8437 | 31.3391 |
| | gemini | 8.6692 | 0.8015 | 1.3627 | 21.6321 | 18.4815 |
| GM | Buy & Hold | 0.3393 | 0.0179 | 2.3823 | 37.8181 | 23.0317 |
| | GPT-4 | 10.5648 | 0.7671 | 1.7351 | 27.5443 | 11.1285 |
| | GPT-4o | -7.0147 | -0.5263 | 1.6792 | 26.6569 | 21.5978 |
| | GPT3.5-Turbo | -17.6385 | -0.9692 | 2.2928 | 36.3976 | 23.0317 |
| | llama2-70B | 8.4911 | **2.6369** | **0.4057** | **6.4402** | **2.1318** |
| | llama3-70B | 25.9335 | 1.9823 | 1.6483 | 26.1657 | 13.2485 |
| | gemini | 18.6257 | 2.4672 | 0.9511 | 15.0989 | 3.0369 |
| NIO | Buy & Hold | -49.4263 | -1.1895 | 5.2351 | 83.1048 | 52.2083 |
| | GPT-4 | **24.7684** | **0.9438** | 3.3063 | 52.4861 | 29.3384 |
| | GPT-4o | -48.3748 | -1.5026 | 4.0562 | 64.3897 | 59.4037 |
| | GPT3.5-Turbo | -28.9321 | -1.0096 | 3.6105 | 57.3149 | 39.5907 |
| | llama2-70B | -49.6947 | -2.7868 | **2.2466** | **35.6639** | 42.6221 |
| | llama3-70B | -28.6668 | -0.7094 | 5.0912 | 80.8202 | 37.1544 |
| | gemini | 14.5673 | 0.6212 | 2.9543 | 46.8977 | **23.0110** |
| COIN | Buy & Hold | -18.4787 | -0.3369 | 6.9098 | 109.6904 | 60.5084 |
| | GPT-4 | 25.7631 | 0.5619 | 5.7761 | 91.6934 | 35.7526 |
| | GPT-4o | -14.2451 | -0.2892 | 6.2049 | 98.4997 | 65.3090 |
| | GPT3.5-Turbo | 25.1141 | 0.4772 | 6.6312 | 105.2669 | 53.9628 |
| | llama2-70B | 15.1836 | 0.4395 | **4.3528** | **69.0979** | **35.3249** |
| | llama3-70B | 19.8876 | 0.3749 | 6.6842 | 106.1076 | 55.7225 |
| | gemini | **89.4782** | **1.7648** | 6.3879 | 101.4048 | 40.3246 |

The FinLLM Challenge is a specialized shared task tailored for LLMs, targeting a comprehensive range of financial problems through three subtasks: financial classification, financial text summarization, and single stock trading. To rigorously evaluate the capabilities of financial LLMs, we have curated three distinct datasets corresponding to each of these subtasks, as detailed in Table 11. This structured approach ensures a holistic and effective assessment of LLM performance across diverse financial scenarios.

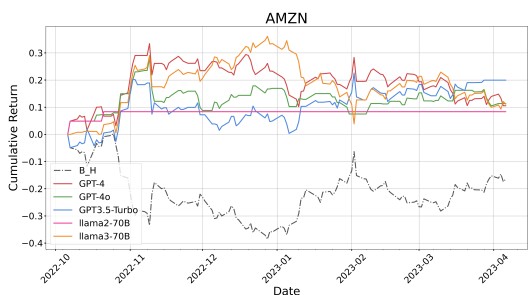

Figure 3: Accumulative Returns of LLM Trading Strategies on AMZN

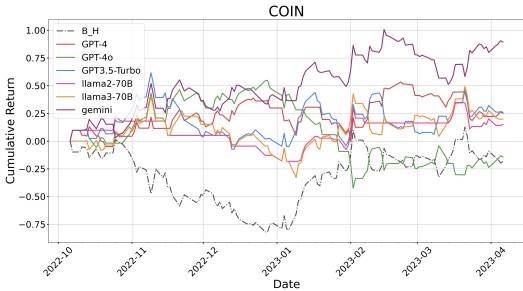

Figure 4: Accumulative Returns of LLM Trading Strategies on COIN

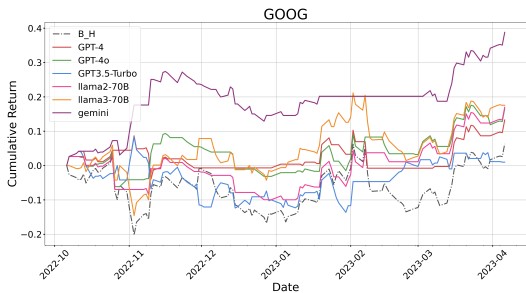

Figure 5: Accumulative Returns of LLM Trading Strategies on GOOG

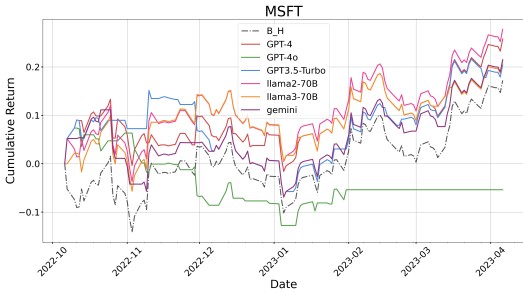

Figure 6: Accumulative Returns of LLM Trading Strategies on MSFT

## G.1 Tasks and Datasets

**Task 1: Financial Classification**. This task, inherited from FinBen's financial classification task, focuses on argument unit classification to test the capabilities of LLMs to identify and categorize texts as premises or claims. It consists of $7.75K$ training data and 969 test data to categorize sentences as claims or premises. We use two metrics to evaluate classification capability, like F1 and Accuracy. F1 score is used as the final ranking metric.

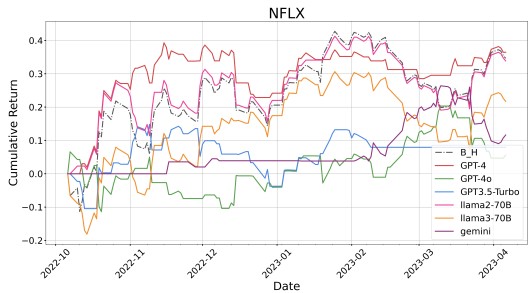

Figure 7: Accumulative Returns of LLM Trading Strategies on NFLX

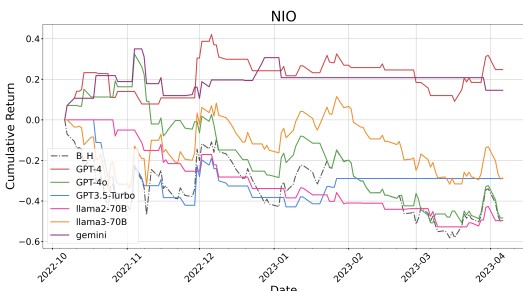

Figure 8: Accumulative Returns of LLM Trading Strategies on NIO

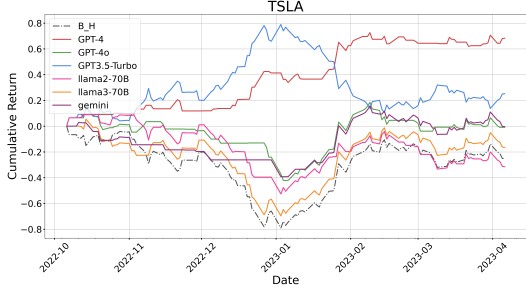

Figure 9: Accumulative Returns of LLM Trading Strategies on TSLA

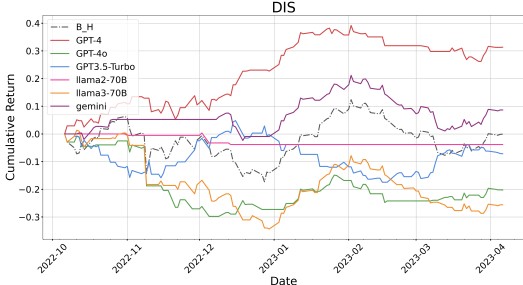

Figure 10: Accumulative Returns of LLM Trading Strategies on DIS

**Task 2: Financial Text Summarization**. This task, inherited from FinBen's generation task, is designed to test the capabilities of LLMs to generate coherent summaries. It provides 8k training data and 2k test data for abstracting financial news articles into concise summaries. We utilize three metrics, such as ROUGE (1, 2, and L) and BERTScore, to evaluate generated summaries in terms of Relevance. ROUGE -1 score is used as the final ranking metric.

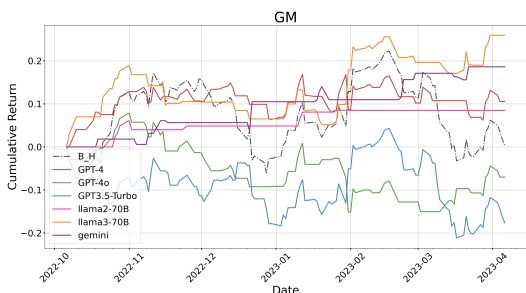

Figure 11: Accumulative Returns of LLM Trading Strategies on GM

Table 11: Tasks and Datasets of FinLLM Challenge.

| Category | Tasks | Datasets | | Evaluation Metrics |
| --- | --- | --- | --- | --- |
| | | Training set | Test set | |
| Task 1 | Financial Classification | 7.75k | 969 | F1 Score, Acc |
| Task 2 | Financial Text Summarization | 8k | 2k | ROUGE-1, ROUGE-2, ROUGE-L, BERTScore |
| Task 3 | Single Stock Trading | 291 | 225 | Sharpe Ratio, Cumulative Return, Maximum Drawdown, Daily and Annualized Volatility, |

**Task 3: Single Stock Trading**. This task, inherited from FinBen's Trading task, aims to evaluate LLMs' ability to make sophisticated decisions in trading activities, which is currently restricted by human's limited ability to process large volumes of data rapidly. It specifically provides 291 data different from FinBen datasets, to evaluate LLMs on sophisticated stock Decisions. We offer a comprehensive assessment of profitability, risk management, and decision-making prowess by a series of metrics, such as Sharpe Ratio (SR), Cumulative Return (CR), Daily (DV) and Annualized volatility (AV), and Maximum Drawdown (MD). Sharpe Ratio (SR) score is used as the final ranking metric.

### G.2 Model Cheating Detection

To measure the risk of data leakage from the test set used in training, we introduce the Data Leakage Test (DLT). The DLT calculates the difference in perplexity between the training set and the test set. A larger difference indicates a lower likelihood of model cheating, while a smaller difference suggests a higher likelihood. For our FinLLM Challenge, we invite Top-3 participant teams per task for cheating detection.

### G.3 Participants and Automatic Evaluation

There are 35 teams registered for FinLLM Challenge, with 12 teams submitting their system description papers. Participants can opt to join one or more task(s).

As shown in Table 12, the top 3 teams achieved outstanding performance in Task 1. Their models' F1 scores were comparable to LlaMA3-8B, although slightly inferior to GPT-4 and LLaMA2-70B, yet significantly outperformed FinMA and other models. The results in Table 12 further demonstrate that our FinLLM share task provides an excellent framework for participating teams to achieve superior experimental outcomes.

Table 12: The Result of Taks 1: Financial Classification

| Teams | ACC | F1 | MCC |
| --- | --- | --- | --- |
| Team Barclays | 0.7626 | 0.5237 | 0.7427 |
| Albatross | 0.7574 | 0.5174 | 0.7555 |
| L3iTC | 0.7544 | 0.5149 | 0.7581 |
| Wealth Guide | 0.7513 | 0.5018 | 0.7406 |
| Finance Wizard | 0.7286 | 0.4554 | 0.7008 |
| CatMemo | 0.711 | 0.4199 | 0.6818 |
| Upaya | 0.709 | 0.4166 | 0.6941 |
| Vidra | 0.7079 | 0.4141 | 0.69 |
| jt | 0.4933 | 0.0141 | 0.5905 |

As illustrated in Table 13, in terms of the Rouge-1 metric, the models of these three teams surpassed all other models, demonstrating superior performance. The results in Table 2 indicate that, for financial generation tasks, our provided dataset and model framework help participating teams leverage their strengths and achieve better outcomes.

Table 13: The Result of Taks 2: Financial Text Summarization

| Teams | Rouge-1 | Rouge-2 | Rouge-L | BertScore | BartScore |
|---|---|---|---|---|---|
| Wealth Guide | 0.308893532 | 0.179468097 | 0.281924302 | 0.85959909 | -4.961457408 |
| Albatross | 0.369077581 | 0.201058395 | 0.322684316 | 0.872049115 | -3.933526929 |
| LBZ | 0.534616211 | 0.358105428 | 0.492179554 | 0.911732047 | -3.407560172 |
| L3iTC | 0.366093426 | 0.187210467 | 0.304610677 | 0.875037043 | -4.257126737 |
| Finance Wizard | 0.521037018 | 0.34060938 | 0.473530112 | 0.90836845 | -3.497988865 |
| Vidra | 0.284955468 | 0.134760859 | 0.228638961 | 0.858682767 | -4.169740305 |
| Revelata | 0.500411369 | 0.333023818 | 0.464356474 | 0.907018743 | -3.805486962 |
| Upaya | 0.529459817 | 0.358203218 | 0.486046685 | 0.910644962 | -3.45155009 |

As shown in Table14, the Top-1 Wealth Guide team excelled in the Sharpe Ratio metric, surpassing other teams and demonstrating outstanding performance. While it may not match the performance of GPT-4, it still outperforms other large models. These results from Table 3 once again underscore the significance of organizing the FinLLM share task. The FinLLM Challenge not only assesses the performance of large language models (LLMs) but also fosters further research into applying LLMs in the financial domain.

Table 14: The Result of Taks 3: Single Stock Trading

| Teams | Sharpe Ratio | Sharpe Ratio-DRIV | Sharpe Ratio-FORM | Sharpe Ratio-JNJ | Sharpe Ratio-MSFT |
|---|---|---|---|---|---|
| Wealth Guide | 0.9263852228 | 0.485625528 | 1.585611423 | 0.078737051 | 1.555566991 |
| Upaya | 0.467489019 | 0.380232272 | 0.108506918 | -1.102831656 | -0.278385232 |
| Albatross | 0.48383204 | 0.251306057 | -1.435471054 | -1.558522674 | 1.309971626 |
| CatMemo | -0.619939784 | -1.393291177 | 0.175932289 | 0.383243051 | -0.879157198 |

# H Performances of non-LLM methods

In this section, we present the performances of non-LLM methods on stock movement prediction and financial NLP tasks from previous papers. Note that non-LLM methods are task-oriented, each model can only run on a specific task.

## H.1 Stock Movement Prediction

Stock movement prediction performance of non-LLM models are shown in Table 15. The results are from (Xie et al., 2023b).

Table 15: Stock movement prediction performance of non-LLM models, measured with the accuracy (ACC) and the Matthews correlation coefficient (MCC). The best performance is in bold.

| Method | BIGDATA22 | | ACL18 | | CIKM18 | |
|---|---|---|---|---|---|---|
| | ACC | MCC | ACC | MCC | ACC | MCC |
| Logistic regression (LR) | 0.53 | 0.02 | 0.52 | 0.04 | 0.53 | -0.04 |
| Random forest (RF) | 0.47 | -0.11 | 0.52 | 0.03 | 0.54 | 0.01 |
| LSTM | 0.51 | 0.01 | 0.53 | 0.06 | 0.53 | 0.02 |
| Attention LSTM (ALSTM) | 0.49 | -0.03 | 0.52 | 0.04 | 0.53 | -0.01 |
| Adv-ALSTM | 0.50 | 0.01 | 0.53 | 0.07 | 0.54 | 0.02 |
| DTML | 0.52 | 0.07 | 0.58 | 0.18 | 0.54 | -0.00 |
| XGBoost | 0.52 | -0.04 | 0.49 | -0.02 | **0.58** | 0.07 |
| XGBRegressor | 0.46 | -0.13 | 0.50 | -0.01 | 0.53 | -0.03 |
| ALSTM-W | 0.48 | -0.01 | 0.53 | 0.08 | 0.54 | 0.03 |
| ALSTM-D | 0.49 | 0.01 | 0.53 | 0.07 | 0.50 | -0.04 |
| StockNet | 0.53 | -0.00 | 0.54 | -0.03 | 0.52 | -0.02 |
| SLOT | **0.55** | **0.10** | **0.59** | **0.21** | 0.56 | **0.09** |

## H.2 Financial NLP Tasks

BERT-based model results of financial NLP tasks are shown in Table 16. The results are from (Shah et al., 2022).

Table 16: Financial NLP tasks performances of BERT-based models. The best performance is in bold.

| Method | FPB Accuracy | Headline AvgF1 | NER F1 | FiQA SA MSE |
|---|---|---|---|---|
| BERT-base | 0.856 | 0.967 | 0.79 | 0.073 |
| FinBERT | 0.872 | 0.968 | 0.8 | 0.070 |
| FLANG-BERT | 0.912 | 0.972 | 0.83 | 0.054 |
| ELECTRA | 0.881 | 0.966 | 0.78 | 0.066 |
| FLANG-ELECTRA | 0.919 | 0.98 | 0.82 | 0.034 |

## H.3 Financial Risk Management Tasks

Traditional model results of financial risk management tasks are shown in Table 17. The results are from (Feng et al., 2024).

Table 17: Performance of various models on financial risk management datasets. The best performance for each metric is in bold.

| Dataset | Method | Metric | Value |
|---|---|---|---|
| Credit Card Fraud | ANN | F1 | 0.85 |
| | | MCC | 0.17 |
| ccfraud | EGRNN++ | F1 | **0.90** |
| | | MCC | **0.34** |
| Polish | Bayesian | F1 | **0.99** |
| | | MCC | **0.57** |
| Travel Insurance | Random Forest | F1 | 0.91 |
| | | MCC | 0.15 |

## Limitations

Despite the novel efforts to benchmark LLMs in the financial domain through FinBen, we acknowledge several inherent limitations that could impact the benchmark's effectiveness and applicability: **Dataset Size Limitations**: The restricted size of available datasets, a common issue in open-source financial data, may affect the models' financial understanding and generalization across various contexts. **Model Size Limitations**: Due to computational constraints, our evaluation was limited to the LLaMA 70B model, potentially overlooking the capabilities of larger or differently architected models. **Generalizability**: The tasks, particularly trading and forecasting, are based on American market data and English texts, possibly limiting the benchmark's applicability to global financial markets. **Potential Negative Impacts**: While FinBen aims to advance financial language understanding, it is crucial to consider potential misuse, such as propagating financial misinformation or exerting unethical influence on markets. Responsible usage and further safeguards are essential[13].

## Ethical Statement

The development and dissemination of the FinBen by the authors carry full responsibility for any potential violation of rights or arising legal issues. All raw data we used are publicly available and do not contain any personal information. Diligent efforts have been undertaken to ensure the construction of the FinBen respects privacy and conforms to established ethical guidelines. The datasets compiled within FinBen are shared under the MIT license, with the expectation that users agree to adhere to its conditions.

This manuscript, inclusive of any associated source codes, datasets, and appendices ("Material"), is designated exclusively for academic and educational pursuits. It is crucial to acknowledge that the Material does not provide financial, legal, or investment counsel, nor should it be utilized as a foundation for any form of decision-making.

While the authors have exerted reasonable diligence to verify the accuracy and reliability of the Material, no explicit or implied warranty is extended regarding its completeness or suitability for any

---

[13]For a detailed ethical and legal statement concerning this work, please see Appendix.

specific application. The authors, along with their affiliated entities, absolve themselves of liability for any losses, damages, or other consequences, whether direct or indirect, that may emanate from the employment or reliance upon the Material. It is incumbent upon the user to seek professional consultation for financial, legal, or investment determinations.

By referencing or employing this Material, individuals consent to indemnify, defend, and hold the authors, along with any affiliated organizations or persons, harmless against any claims or damages that may arise from such utilization.

**Disclaimer: We are sharing codes for academic purposes under open-source license. Nothing herein is financial advice, and NOT a recommendation to trade real money. Please use common sense and always first consult a professional before trading or investing.**

