# A    Additional Results

The performance analysis of other LLMs on the acronym and regulations tasks, as shown in Tables 1 and 2, provides valuable insights into their capabilities.

The acronym dataset is a QA task that requires models to decode financial acronyms. Despite not having seen this task before, FinMA, a financial LLM specially trained on financial tasks, performed exceptionally well. The FinMA7B-full model achieved the highest ROUGE-1 score of 0.12 and the highest BERTScore of 0.73, even surpassing GPT-4. This indicates that financial-specific models can leverage their domain knowledge effectively, even on short QA tasks like the acronym dataset.

On the other hand, the regulations dataset involves answering intricate questions related to financial regulations, such as EMIR. This task is long, complex, and difficult to understand, posing a significant challenge. In this scenario, the LLaMA2-70b-chat model stand out with a ROUGE-1 score of 0.30 and a BERTScore of 0.68, highlighting its ability to handle complex regulatory questions. This underscores the importance of model size and capability when dealing with more demanding and sophisticated tasks in the financial domain.

Table 1: Results of acronym dataset. The best performance is in bold.

| | LLaMA3 -8B | LLaMA2 -7B-hf | FinMA7B -full | InternLM -chat-7B | FinGPT -mt | Falcon -7B | cfGPT | Baichuan -7B | CodeLLaMA | DISC -FinLLM | ChatGLM3 -6b | LLaMA2 -70b-chat | Mistral-7B -instruct |
|---|---|---|---|---|---|---|---|---|---|---|---|---|---|
| ROUGE-1 | 0.02 | 0.03 | **0.12** | 0.10 | 0 | 0.01 | 0.05 | 0 | 0.01 | 0.02 | 0.04 | 0.02 | 0.12 |
| BERTScore | 0.55 | 0.57 | **0.73** | 0.68 | 0.64 | 0.54 | 0.69 | 0.48 | 0.45 | 0.57 | 0.62 | 0.58 | 0.72 |

Table 2: Results of regulations dataset. The best performance is in bold.

| | LLaMA3 -8B | LLaMA2 -7B-hf | FinMA7B -full | InternLM -chat-7B | FinGPT -mt | Falcon -7B | cfGPT | Baichuan -7B | CodeLLaMA | DISC -FinLLM | ChatGLM3 -6b | LLaMA2 -70b-chat | Mistral-7B -instruct |
|---|---|---|---|---|---|---|---|---|---|---|---|---|---|
| ROUGE-1 | 0.10 | 0.24 | 0.12 | 0.04 | 0.01 | 0.03 | 0.14 | 0.13 | 0.17 | 0.12 | 0.26 | **0.30** | 0.28 |
| BERTScore | 0.60 | 0.65 | 0.59 | 0.57 | 0.40 | 0.14 | 0.57 | 0.60 | 0.59 | 0.52 | 0.65 | **0.68** | 0.67 |

# B    Motivation For Datasheet Creation

## B.1    Why was the datasheet created? (e.g., was there a specific task in mind? was there a specific gap that needed to be filled?)

FinBen was created to address the gap in comprehensive benchmarks and evaluation studies of large language models within the financial domain. Despite the proven capabilities of LLMs such as GPT-4 in transforming various fields including finance, a detailed understanding of their potential and limitations specific to finance is still lacking. This is partly due to the complex and specialized nature of financial tasks, which necessitates targeted datasets for thorough analysis. By evaluating 42 datasets covering 24 financial tasks, we aim to provide a robust benchmark that allows researchers and practitioners to evaluate the effectiveness of LLMs in financial text analysis and prediction tasks more accurately and reliably. These datasets are thus a critical step towards leveraging the full capabilities of LLMs in the finance sector, ensuring that their deployment is both effective and appropriate for the intricacies of financial applications.

## B.2    Has the dataset been used already? If so, where are the results so others can compare (e.g., links to published papers)?

Yes, the dataset has already been used. It was employed in the FinLLM Share Task during the FinNLP-AgentScen Workshop at IJCAI 2024, known as the FinLLM Challenge. This event saw active participation, with 35 teams registering to take part and 12 of those teams submitting system description papers. This indicates that datasets of FinBen, have been utilized to test and explore the capabilities of LLMs within the financial sector, thereby promoting further research and understanding in this area. The link of the challenge is `https://sites.google.com/nlg.csie.ntu.edu.tw/finnlp-agentscen/shared-task-finllm?authuser=0`.

### B.3 What (other) tasks could the dataset be used for?

The 42 datasets of FinBen, can be employed for several tasks within the financial technology (FinTech) sector, covering seven critical aspects: information extraction (IE), textual analysis, question answering (QA), text generation, risk management, forecasting, decision-making, and Spanish. It can be utilized to train and evaluate LLMs for a variety of applications, including:

**Financial Sentiment Analysis**: Analyzing sentiments in financial texts such as market news, analyst reports, and social media to gauge investor sentiment and predict market trends.

**Fraud Detection**: Enhancing models that detect fraudulent activities by analyzing transaction patterns and communication within financial institutions.

**Financial Forecasting**: Improving predictions of financial markets, stock prices, economic indicators, or company performance metrics based on historical data and current market conditions.

**Personalized Financial Advice**: Generating customized financial advice for individuals based on their spending habits, investment preferences, and financial goals.

**Regulatory Compliance**: Assisting in compliance monitoring by analyzing communications and transactions to ensure they meet legal and regulatory standards.

These applications demonstrate the potential of the FinBen dataset to significantly impact various aspects of financial technology by providing robust training and evaluation grounds for sophisticated LLMs tailored to specific financial tasks.

### B.4 Who funded the creation dataset?

FinBen is a collaborative project carried out by researchers from several institutions, including Wuhan University, the University of Manchester, University of Florida, Columbia University, The Chinese University of Hong Kong, Shenzhen, Sichuan University, Yunnan University, Stevens Institute of Technology, Stony Brook University, Nanjing Audit University, Jiangxi Normal University, Southwest Jiaotong University. The project is funded by the Fin AI.

### B.5 Any other comment?

FinBen represents a significant breakthrough in the domain of financial AI. It aims to bridge the gap in the lack of open-source financial benchmarks for evaluating LLMs and providing comprehensive datasets and evaluation metrics for these tasks. It is meticulously designed to enhance the performance of financial LLMs in downstream tasks by providing a novel taxonomy for organizing financial evaluation tasks and systematic evaluation metrics.

Moreover, FinBen is designed to foster transparency and collaboration in the field. It openly provides instruction-tuning data, evaluation datasets, and a structured framework included in the benchmark to encourage open research. By covering a diverse set of financial tasks, FinBen offers a versatile tool for both academic researchers and industry professionals.

By providing these resources and promoting open-source development, FinBen is set to push forward the frontier of financial AI, offering a comprehensive framework for assessing financial LLMs and fostering a deeper understanding of complex financial language and concepts. This project is a significant step towards the integration of advanced AI techniques into the financial industry, paving the way for a wide range of applications such as predicting stock price movements and advanced financial analytics.

## C Datasheet Composition

## C.1 What are the instances?(that is, examples; e.g., documents, images, people, countries) Are there multiple types of instances? (e.g., movies, users, ratings; people, interactions between them; nodes, edges)

The instruction dataset within the FinBen benchmark comprises individual samples of data that are used to evaluate the model. These instances are varied and encompass multiple types, reflecting the diverse tasks and data sources prevalent in the financial domain.For instance, in the named entity recognition task, an example would involve a sentence from a financial agreement, where the model is instructed to identify named entities corresponding to a person, organization, or location.

The data types include:

- Textual Data: Predominantly, the instructions pertain to processing textual data such as reports, news articles, news headlines, regulatory filings, tweets, and financial agreements. Tasks associated with this data type include sentiment analysis, news headline classification, and named entity recognition.

- Time-Series Data: Instructions sometimes relate to time-series data, specifically stock price data, often used in conjunction with textual data for tasks like stock movement prediction.

- Tabular Data: Instructions also cover the processing of tabular data, typically derived from companies' financial filings. This data is essential for question-answering tasks where the model needs to extract and leverage information from financial tables.

## C.2 How many instances are there in total (of each type, if appropriate)?

The test dataset is broken down by each task type, with the total number of instances for each as follows:

- Sentiment Analysis and Related Tasks: 4,619 instances
- Stock Movement Prediction: 6,330 instances
- Question Answering and Related Tasks: 2,305 instances
- Named Entity Recognition and Related Tasks: 3,228 instances
- Credit Scoring and Fraud Detection: 6,461 instances
- Summarization: 990 instances
- Trading: 3,384 instances
- Spanish: 3,025 instances

This results in a total of 30,342 instances in the test dataset.

## C.3 What data does each instance consist of ? "Raw" data (e.g., unprocessed text or images)? Features/attributes? Is there a label/target associated with instances? If the instances related to people, are subpopulations identified (e.g., by age, gender, etc.) and what is their distribution?

- Information Extraction: Instances are sourced from various financial documents like SEC filings and financial agreements. They include tasks such as named entity recognition, relation extraction, causal classification, and detection, numeric labeling, and textual analogy parsing. These tasks require the model to identify and classify text spans, relationships, and numeric data based on specific instructions.

- Textual Analysis: This includes tasks like sentiment analysis, news headline classification, and various forms of argument and claim classification. Instances here are mostly textual data from financial news, earnings reports, and policy documents. They are structured to help models analyze sentiments, classify text into predefined categories, or extract argumentative structures.

- Question Answering: Instances include financial reports and tables, challenging models to perform tasks such as multi-step numerical reasoning and multi-turn question answering. These are designed to test a model's ability to navigate complex financial data and provide accurate, context-aware responses.
- Text Generation: Focuses on generating coherent summaries from financial texts like earnings call transcripts and news articles. Instances are tailored to test the model's capabilities in condensing information while retaining critical financial insights and facts.
- Forecasting and Risk Management: These instances involve predicting future financial events or assessing risks from data like historical stock prices and customer transactions. Tasks include stock movement prediction, credit scoring, fraud detection, and financial distress identification.
- Decision-making: Strategic decision-making instances challenge models to synthesize information for trading strategies. They include simulated trading environments with historical price data and market sentiment, designed to evaluate the profitability and risk management capabilities of financial LLMs.
- Spanish: Instances include news, company fillings, and examninations in Spanish, which aim to assess the Spanish ability of financial LLMs.

### C.4 Is there a label or target associated with each instance? If so, please provide a description.

Each task in the dataset has a specific target or label associated with it, as described below:
- Information Extraction: Labels identify entities, relationships, causal links, numeric values, or roles within analogies.
- Textual Analysis: Targets include sentiment classifications, news relevance, argument stances, and categorization of financial texts.
- Question Answering: Answers or information extracted from financial documents and tables serve as the targets.
- Text Generation: Summaries or generated texts are evaluated against reference summaries to assess quality and accuracy.
- Forecasting and Risk Management: Predicted outcomes, risk categories, or fraud indicators are the labels for these tasks.
- Decision-making: Performance metrics such as returns and risk ratios are the targets in simulated trading scenarios.
- Spanish: Targets include sentiment classifications, categorization of financial texts, summaries in Spanish.

### C.5 Is any information missing from individual instances? If so, please provide a description, explaining why this information is missing (e.g., because it was unavailable). This does not include intentionally removed information, but might include, e.g., redacted text.

In the assembly and preprocessing stages of our current dataset, we have aimed to maintain a comprehensive dataset for each task. Despite these efforts, there are instances where certain information might be missing due to several factors:
- **Unavailability:** At times, specific data might not be accessible during collection. For instance, certain financial reports or articles may lack detailed information about company performance or proprietary data.
- **Irrelevance:** Occasionally, certain details are deliberately excluded because they do not aid the task objectives. For example, the omission of 'MISCELLANEOUS' entities in Named Entity Recognition tasks, as these do not help in identifying key entities like persons, organizations, or locations.

- **Privacy Protection:** In tasks involving stock movement prediction, tweets are sanitized to remove any personally identifiable information or offensive content, adhering to privacy regulations and maintaining ethical research standards.
- **Token Limitation:** For tasks such as ConvFinQA, FinQA, and stock price movement prediction, the length of instances is capped. Any data exceeding 2048 tokens is truncated due to the sequence length limitations of LLMs. This truncation is carefully managed to retain the most pertinent information for the task, ensuring that crucial context (e.g., relevant financial news or the specific question context) remains within the token limit.

## C.6  Are relationships between individual instances made explicit (e.g., users' movie ratings, social network links)? If so, please describe how these relationships are made explicit.

In our dataset, each entry is treated as an individual instance with a primary focus on specific tasks such as sentiment analysis, named entity recognition, question answering, and others. Relationships between individual instances are generally not explicitly defined, but certain inherent connections may exist:

- **Temporal Relationships:** In tasks like stock movement prediction, instances are connected through their temporal sequence. For example, stock prices and related tweets follow a chronological order, creating an implicit link based on their timestamps.
- **Contextual Relationships:** For question answering tasks, such as ConvFinQA and FinQA, questions and their answers may be interconnected as part of a continuous conversation or discussion thread.

It is crucial to recognize that while these relationships are inherent to the data, they are not explicitly labeled or annotated within the dataset. The model is expected to discern and utilize these relationships during its training process.

## C.7  Does the dataset contain all possible instances or is it a sample (not necessarily random) of instances from a larger set? If the dataset is a sample, then what is the larger set? Is the sample representative of the larger set (e.g., geographic coverage)? If so, please describe how this representativeness was validated/verified. If it is not representative of the larger set, please describe why not (e.g., to cover a more diverse range of instances, because instances were withheld or unavailable).

The dataset is a sample drawn from a larger set of financial data sources, including financial reports, news articles, stock prices, tweets, and other financial and economic texts. Due to the sheer volume and continuously evolving nature of such data, the dataset does not cover all possible instances from these sources. The samples were selected to enable diverse and comprehensive coverage of typical tasks in the financial domain.

- **Representativeness:** The dataset strives to be representative of the larger set of financial data sources in terms of the variety of tasks it covers (e.g., sentiment analysis, entity recognition, question answering, stock movement prediction), modalities (text, time-series data), and types of financial texts (news articles, reports, regulatory filings).
- **Validation:** Due to the dynamic and vast nature of the larger data set, the representativeness of the dataset cannot be fully validated. However, it was designed to include diverse and important tasks in financial analysis, and its effectiveness is evaluated based on the performance of models trained on it.
- **Limitations:** Some instances may have been excluded due to data unavailability, restrictions in data sharing agreements, or practical considerations such as token limitations in the model. For example, very lengthy reports or conversations may have been truncated or excluded. Additionally, some types of financial data (e.g., confidential company reports, private communications) are inherently unavailable due to privacy and confidentiality reasons.

Despite these limitations, the dataset aims to provide wide coverage of tasks, modalities, and text types, capturing the complexity and diversity of financial data analysis tasks.

### C.8  Are there recommended data splits (e.g., training, development/validation, testing)? If so, please provide a description of these splits, explaining the rationale behind them.

For our benchmark, we have opted to include only a test split to specifically evaluate the performance of large language models (LLMs). This ensures the evaluation focuses solely on the model's ability to generalize to unseen data, a critical aspect for assessing LLM effectiveness. By excluding training and validation splits, we provide a standardized evaluation framework emphasizing the model's performance in real-world scenarios with new and unseen data. For existing datasets, we use only their test sets, and for our newly created datasets, all data is designated for testing.

### C.9  Are there any errors, sources of noise, or redundancies in the dataset? If so, please provide a description.

Given the diverse sources of data and the complex nature of financial language, there may be inherent errors, noise, or redundancies in the dataset, including:

- Sentiment Ambiguity: In the Financial Sentiment Analysis task, converting sentiment scores into three categories (negative, neutral, positive) can introduce noise. This is because sentiments expressed in financial texts are often subjective and ambiguous.
- Named Entity Recognition Noise: The process of discarding miscellaneous entities in the Named Entity Recognition task might inadvertently remove useful information. Additionally, sentences without any entities might introduce further noise.
- Question Answering Complexity: In the Question Answering task, the complexity of financial questions and their respective answers may lead to potential errors during annotation. Multiple correct or partially correct answers might exist for a given question, adding to the challenge.
- Stock Movement Prediction: In Stock Movement Prediction, price features and tweet data might contain redundancies, as they include multiple overlapping features such as opening, highest, lowest, closing, and adjusted closing prices. Moreover, stock movement predictions are influenced by various factors, not all of which may be captured in the features used, leading to potential errors.
- Token Limit: For tasks such as ConvFinQA, FinQA, and Stock Movement Prediction, instances are truncated to 2048 tokens, which might result in the loss of information and subsequently introduce errors in these tasks.

Despite these potential issues, the datasets remain valuable resources for developing and benchmarking models for financial tasks, provided these limitations are considered during the model development and evaluation process.

### C.10  Is the dataset self-contained, or does it link to or otherwise rely on external resources (e.g., websites, tweets, other datasets)? If it links to or relies on external resources, a) are there guarantees that they will exist, and remain constant, over time; b) are there official archival versions of the complete dataset (i.e., including the external resources as they existed at the time the dataset was created); c) are there any restrictions (e.g., licenses, fees) associated with any of the external resources that might apply to a future user? Please provide descriptions of all external resources and any restrictions associated with them, as well as links or other access points, as appropriate.

The dataset originates from external sources like financial news articles, tweets, and stock market data, but it has been preprocessed, labeled, and stored independently, making it self-contained. This ensures no direct links to the original sources, mitigating issues of persistence, archival consistency, and restrictions.

Hosted on a GitHub repository, the dataset is easily accessible and benefits from GitHub's version-control features, preserving its state at release. Users can clone or fork the repository under the specified terms and conditions.

Despite its external origins, the final dataset is self-contained, accessible, and minimally restricted.

Any other comments?

The dataset for each task balances comprehensiveness and specificity well. Key points include:

- Diversity: The dataset includes diverse text data from various sources, ensuring broad coverage of financial terms and expressions, particularly in sentiment analysis and news headline classification tasks.
- Representation: It represents real-world financial NLP challenges, with instances carefully chosen to simulate tasks like sentiment analysis, entity recognition, question answering, and stock movement prediction.
- Comprehensiveness: Covering a wide range of tasks from text classification to complex question answering and stock movement prediction, the dataset addresses various financial NLP problems.
- Size: It is sufficiently large, providing ample instances for training, validation, and testing robust models.

These factors enhance the dataset's utility and value for financial LLMs.

## D    Collection Process

### D.1    What mechanisms or procedures were used to collect the data (e.g., hardware apparatus or sensor, manual human curation, software program, software API)? How were these mechanisms or procedures validated?

The dataset has been assembled from publicly available datasets and resources, supplemented by the manual creation of instruction templates. More specifically:

- **Publicly Available Datasets:** Existing datasets have been utilized in our tasks, including financial sentiment analysis, news headline classification, named entity recognition and question answering. These datasets are widely acknowledged and utilized in the research community, validating their reliability and quality. The raw datasets are provided along with our work for reference.
- **Manual Creation of Instruction Templates:** For tasks that rely on conversational models, we have manually created instruction templates. These templates guide the model's responses and ensure that the output is consistent with the task's requirements.

The full details, including the source of the raw datasets, are provided in the documentation accompanying our GitHub repository. The use of publicly available datasets, coupled with manual curation of instructions, helps ensure the data's reliability and robustness for our tasks.

### D.2    How was the data associated with each instance acquired? Was the data directly observable (e.g., raw text, movie ratings), reported by subjects (e.g., survey responses), or indirectly inferred/derived from other data (e.g., part-of-speech tags, model-based guesses for age or language)? If data was reported by subjects or indirectly inferred/derived from other data, was the data validated/verified? If so, please describe how.

We constructed the multi-task and multi-modal instruction data by collecting publicly available training data from a range of diverse tasks. This data was directly observable and derived from multiple open-released financial datasets. The data we used included both textual and time-series

data modalities, which allowed our model to handle tasks such as sentiment analysis, news headline classification, named entity recognition, question answering, and stock movement prediction. For each task, we wrote specific instructions that were combined with the data samples to create our large-scale instruction tuning data.

The instructions for each task were carefully designed by domain experts to ensure that they accurately reflect the nuances and requirements of the different tasks. This approach allowed us to tailor our large language model, FinMA, to perform a diverse range of financial tasks.

### D.3 If the dataset is a sample from a larger set, what was the sampling strategy (e.g., deterministic, probabilistic with specific sampling probabilities)?

Our dataset is not a sample from a larger dataset but an assembly of publicly available multi-task and multi-modal data derived from multiple open-released financial datasets. We didn't use a specific sampling strategy since we weren't sampling from a larger set. Instead, we collected the complete datasets that were available and relevant to our study

### D.4 Who was involved in the data collection process (e.g., students, crowd workers, contractors) and how were they compensated (e.g., how much were crowd workers paid)?

For the FinBen project, our data collection process involved both domain experts and students. The domain experts, who are professionals in the financial field such as fund managers, were invited to design the task-specific instructions for each dataset. On the other hand, students were responsible for the collection of publicly available datasets.

This project was conducted as collaborative research, so those who contributed to the data collection and instruction design were not compensated in a traditional paid manner. Instead, they contributed their expertise and time to the project as part of their research activities or professional engagement.

### D.5 Over what timeframe was the data collected? Does this timeframe match the creation timeframe of the data associated with the instances (e.g., a recent crawl of old news articles)? If not, please describe the timeframe in which the data associated with the instances was created.

- FPB, while it does not provide a specific timeframe for data collection, was published in 2014.
- FIQASA, like FPB, does not specify a timeframe for data collection, but it was published in 2018.
- The Headlines dataset contains human-annotated news headlines that were collected over a span of 19 years, from 2000 to 2019. It was published in 2021.
- NER, again, does not indicate a specific timeframe for data collection, but it was published in 2015.
- The FinQA dataset spans two decades, collecting data from 1999 to 2019. It was published in 2021.
- ConvFinQA, like FinQA, covers a period from 1999 to 2019 and was published in 2022.
- The BigData22 dataset has a more confined range of data collection, from July 5th, 2019 to June 30th, 2020.
- The ACL18 dataset spans approximately two years, from January 2nd, 2014 to December 30th, 2015.
- The CIKM18 dataset covers data collected within a single year, from January 3rd, 2017 to December 28th, 2017.
- FOMC, with the meeting minutes and speeches data spanned from January 1st, 1996 to October 15th, 2022, and the press conferences data from April 27th, 2011 to October 15th, 2022, was published in 2023.

- FinArg-ACC does not specify a timeframe for data collection, but it was published in 2023.
- FinArg-ARC does not specify a timeframe for data collection, but it was published in 2023.
- MultiFin, covering a period from 2015 to 2021, was published in 2023.
- The MA dataset covers data collected from January 1st, 2007 to August 12th, 2019, and was published in 2020.
- MLESG, again, does not specify a timeframe for data collection, but it was published in 2023.
- FiNER-ORD does not specify a timeframe for data collection, but it was published in 2023.
- FinRED collected the raw data during the timespan of 2015,2019, respectively. it was published in 2023.
- SC extracted data during 2019 and was published in 2020.
- CD extracted data during 2019 and was published in 2020.
- TATQA collected data from 2019 to 2020 and was published in 2021.
- The ECTSum dataset spans over three years, from January 2019 to April 2022.
- EDTSum does not specify a timeframe for data collection, but it was published in 2021.
- The fintrade dataset spans approximately two years, from August 15, 2021, to April 25, 2023. it was published in 2023.
- FNXL covering a period from 2019 to 2021, was published in 2023.
- For Spanish datasets, please refer to **?**.

In summary, the data associated with the instances in the FinBen dataset spans a wide range of timeframes, from specific periods within a year to a stretch of 20 years. Some of these match the publication years of their respective sources, like the Headlines dataset, FinQA, and ConvFinQA, while others, like FPB, FIQASA, and NER, do not specify a timeframe for data collection. Therefore, the creation timeframe of the data associated with the instances varies, and in some cases, it may be assumed to be close to their respective publication years.

## E   Data Preprocessing

**E.1  Was any preprocessing/cleaning/labeling of the data done (e.g., discretization or bucketing, tokenization, part-of-speech tagging, SIFT feature extraction, removal of instances, processing of missing values)? If so, please provide a description. If not, you may skip the remainder of the questions in this section.**

To convert raw data into a structured instruction dataset, we defined a pipeline with the following steps:

1. **Define Instruction Templates:** We began by defining clear and succinct instruction templates for each task.
2. **Extract Relevant Information:** The next step involved extracting the requisite pieces of information from the raw data. These would be used to fill in the instruction templates.
3. **Match to Template:** Once the relevant information was extracted, it was matched to the instruction template.
4. **Verify and Clean Instructions:** Following the creation of the instructions, we verified that they were both coherent and accurate representations of the task.
5. **Standardize Instructions:** Finally, we standardized the instructions in terms of language, structure, and style. This ensured consistency across the dataset, which subsequently enabled the model to understand and learn the tasks more efficiently.

This pipeline was utilized to process each task within the FinBen model. For example, consider a task involving sentiment analysis. The raw data consisted of a series of tweets along with their associated sentiment labels. The instruction template for this task could be: "Classify the sentiment of the following tweet as 'positive', 'negative', or 'neutral'". The preprocessing steps involved extracting the tweet text, fitting it into the template, and verifying that the resulting instruction accurately represented the task.

### E.2 Was the "raw" data saved in addition to the preprocessed/cleaned/labeled data (e.g., to support unanticipated future uses)? If so, please provide a link or other access point to the "raw" data.

The raw data can be accessed through the original papers:

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

32. For Spanish datasets, please refer to **?**.

### E.3 Is the software used to preprocess/clean/label the instances available? If so, please provide a link or other access point.

Yes, the software used for preprocessing, cleaning, and labeling the instances is publicly available. It can be accessed via our GitHub repository at https://github.com/The-FinAI/PIXIU.

### E.4 Does this dataset collection/processing procedure achieve the motivation for creating the dataset stated in the first section of this datasheet? If not, what are the limitations?

Yes, the dataset collection and processing procedure does fulfill the initial motivation. We have created a comprehensive set of resources specifically tailored for the financial sector. However, as with any novel project, there are limitations. The financial domain is vast and complex, and while FinBen covers a broad range of topics, it may not encompass all possible financial tasks or scenarios. We hope that future iterations of FinBen can further expand and diversify the instruction dataset and evaluation benchmarks.

### E.5 Any other comments

No.

## F Dataset Distribution

### F.1 How will the dataset be distributed? (e.g., tarball on website, API, GitHub; does the data have a DOI and is it archived redundantly?)

The dataset, along with the necessary code, will be distributed through our GitHub repository (`https://github.com/The-FinAI/PIXIU`).

### F.2 When will the dataset be released/first distributed? What license (if any) is it distributed under?

The dataset was first released on February 20th, 2024. It is distributed under the MIT License, which permits use, copy, modify, merge, publish, distribute, sublicense, and/or sell copies of the Software

with few restrictions, provided that the above copyright notice and this permission notice are included in all copies or substantial portions of the Software.

### F.3 Are there any copyrights on the data?

The transformed and aggregated dataset, as well as the code and models we provide, are released under the MIT License.

### F.4 Are there any fees or access/export restrictions?

There are no fees associated with the FinBen, and it can be freely accessed and exported for research purposes.

### F.5 Any other comments?

We encourage researchers to use this dataset and related resources to advance the field of financial AI. However, proper attribution should be given when using these resources in line with standard academic practices.

## G Dataset Maintenance

### G.1 Who is supporting/hosting/maintaining the dataset?

The dataset is maintained and supported by a team of researchers from various institutions.

The team includes:

Qianqian Xie[a], Weiguang Han[b], Zhengyu Chen[b], Ruoyu Xiang[a], Xiao Zhang[a], Yueru He[a], Mengxi Xiao[b], Dong Li[b], Yongfu Dai[g], Duanyu Feng[g], Yijing Xu[a], Haoqiang Kang[e], Ziyan Kuang[l], Chenhan Yuan[c], Kailai Yang[c], Zheheng Luo[c], Tianlin Zhang[c], Zhiwei Liu[c], Guojun Xiong[j], Zhiyang Deng[i], Yuechen Jiang[i], Zhiyuan Yao[i], Haohang Li[i], Yangyang Yu[i], Gang Hu[h], Jiajia Huang[k], Xiao-Yang Liu[e], Alejandro Lopez-Lira[d], Benyou Wang[f], Yanzhao Lai[m], Hao Wang[g], Min Peng[b], Sophia Ananiadou[c], Jimin Huang[a].

And the indicators for institutions are:

[a]The Fin AI, [b]Wuhan University, [c]The University of Manchester, [d]University of Florida, [e]Columbia University, [f]The Chinese University of Hong Kong, Shenzhen, [g]Sichuan University, [h]Yunnan University, [i]Stevens Institute of Technology [j]Stony Brook University, [k]Nanjing Audit University, [l]Jiangxi Normal University, [m]Southwest Jiaotong University.

### G.2 Will the dataset be updated? If so, how often and by whom?

The updates will be managed by Zhengyu Chen from the School of Computer Science, Wuhan University. The frequency of updates will be determined by the availability of new data and feedback from users.

### G.3 How will updates be communicated? (e.g., mailing list, GitHub)

Updates about the dataset will be communicated through GitHub, where the dataset is hosted. Users are encouraged to keep track of the GitHub repository for any updates.

### G.4 If the dataset becomes obsolete how will this be communicated?

If the dataset becomes obsolete, this will be communicated through the same GitHub repository where the dataset is hosted.

### G.5 Is there a repository to link to any/all papers/systems that use this dataset?

Currently, there is no specific repository for linking to papers or systems that use the dataset. However, users are encouraged to cite the FinBen paper in their work.

### G.6 If others want to extend/augment/build on this dataset, is there a mechanism for them to do so? If so, is there a process for tracking/assessing the quality of those contributions? What is the process for communicating/distributing these contributions to users?

The dataset is open source and users are welcome to extend, augment, or build on it. Users can submit pull requests to the GitHub repository where the dataset is hosted. The quality of contributions will be assessed by the maintenance team before being integrated into the dataset. These contributions, once approved, will be made available to other users through the GitHub repository.

## H   Legal and Ethical Considerations

### H.1 Were any ethical review processes conducted (e.g., by an institutional review board)? If so, please provide a description of these review processes, including the outcomes, as well as a link or other access point to any supporting documentation.

No specific ethical review processes were conducted for the development of FinBen as it leverages previously published and publicly accessible datasets. These original datasets underwent their own ethical review processes.

### H.2 Does the dataset contain data that might be considered confidential (e.g., data that is protected by legal privilege or by doctor-patient confidentiality, data that includes the content of individuals non-public communications)? If so, please provide a description.

The datasets used in FinBen do not contain any data that might be considered confidential. Any potential personal information in the original datasets has been removed by the original authors before they were made publicly accessible.

### H.3 Does the dataset contain data that, if viewed directly, might be offensive, insulting, threatening, or might otherwise cause anxiety? If so, please describe why

The datasets used in FinBen do not contain data that, if viewed directly, might be offensive, insulting, threatening, or might otherwise cause anxiety. The original authors of the datasets ensured this during their preprocessing and cleaning stages.

### H.4 Does the dataset relate to people? If not, you may skip the remaining questions in this section.

While some of the datasets may originally have pertained to people, for instance, those containing tweets, any direct identifiers have been removed from the original datasets. Therefore, FinBen does not directly relate to identifiable individuals.

**H.5**   **Does the dataset identify any subpopulations (e.g., by age, gender)? If so, please describe how these subpopulations are identified and provide a description of their respective distributions within the dataset.**

No.

**H.6**   **Is it possible to identify individuals (i.e., one or more natural persons), either directly or indirectly (i.e., in combination with other data) from the dataset? If so, please describe how.**

It is not possible to identify individuals either directly or indirectly from the datasets used in FinBen. All potentially identifiable information was removed from the original datasets before they were made publicly accessible.

**H.7**   **Does the dataset contain data that might be considered sensitive in any way (e.g., data that reveals racial or ethnic origins, sexual orientations, religious beliefs, political opinions or union memberships, or locations; financial or health data; biometric or genetic data; forms of government identification, such as social security numbers; criminal history)? If so, please provide a description.**

The datasets used in FinBen do not contain data that might be considered sensitive in any way. All potentially sensitive information was removed from the original datasets before they were made publicly accessible.

**H.8**   **Did you collect the data from the individuals in question directly, or obtain it via third parties or other sources (e.g., websites)?**

The data used in FinBen was not directly collected from individuals. Instead, it was obtained from previously published datasets that were collected by other researchers and made publicly available.

**H.9**   **Were the individuals in question notified about the data collection? If so, please describe (or show with screenshots or other information) how notice was provided, and provide a link or other access point to, or otherwise reproduce, the exact language of the notification itself.**

As the data used in FinBen were obtained from previously published datasets, the original data collectors were responsible for notifying individuals about the data collection.

**H.10**   **Did the individuals in question consent to the collection and use of their data? If so, please describe (or show with screenshots or other information) how consent was requested and provided, and provide a link or other access point to, or otherwise reproduce, the exact language to which the individuals consented.**

As the data used in FinBen were obtained from previously published datasets, the original data collectors were responsible for obtaining consent from individuals for the collection and use of their data.

**H.11**   **If consent was obtained, were the consenting individuals provided with a mechanism to revoke their consent in the future or for certain uses? If so, please provide a description, as well as a link or other access point to the mechanism (if appropriate).**

The issue of revoking consent does not directly apply to FinBen as the data used was obtained from previously published datasets. The original data collectors were responsible for providing mechanisms for revoking consent.

Given that the data used in FinBen is de-identified and does not contain sensitive information, it is not anticipated that there will be direct impacts on data subjects. However, as with any research involving human-related data, there is always a responsibility to use the data ethically and with respect to potential implications.

**H.13 Any other comments?**

No.

# I Responsibility Statement

The authors of FinBen bear all responsibility in case of any violation of rights or any other legal issues that arise from the use of this dataset. The authors have taken all possible measures to ensure the respect of privacy and ethical guidelines in the construction of this dataset.

The datasets included in FinBen are distributed under the MIT license. By using these datasets, users agree to comply with the terms of this license.

This paper, including any associated source codes, datasets, and appendices ("Material"), is intended solely for academic and educational purposes. The Material does not constitute financial, legal, or investment advice and is not intended to be a basis for any decision-making.

While the authors have taken reasonable measures to ensure the accuracy of the Material, no warranty, express or implied, is made as to its completeness, reliability, or suitability for any specific purpose. The authors and their affiliated organizations shall not be liable for any losses, damages, or consequences, whether direct or indirect, arising from the use or reliance on the Material. It is the responsibility of the user to consult with professionals for any financial, legal, or investment decisions.

By referencing or utilizing this Material, the reader agrees to indemnify, defend, and hold harmless the authors and any affiliated organizations or persons from any and all claims or damages arising from such use.

# J Reproducibility

Ensuring the reproducibility of research results is of utmost importance in promoting transparency and enabling further scientific advancements. In the context of the benchmarks presented in this datasheet, we have taken several measures to facilitate the reproducibility of our reported results.

To begin with, we have made all the necessary code, data, and instructions available in the Fin-Ben GitHub repository (`https://github.com/The-FinAI/PIXIU`). This repository serves as a centralized hub where researchers can access the resources required to reproduce the benchmarks. The repository is organized and well-documented, providing a clear and structured framework for replication.

In order to enhance reproducibility, we have adhered to the ML reproducibility checklist, a framework that promotes best practices for ensuring the replicability of machine learning experiments. By following this checklist, we have prioritized the inclusion of all essential components, such as code, datasets, and evaluation procedures, making it easier for researchers to reproduce our reported results.

Furthermore, we have provided detailed instructions within the repository, outlining the steps needed to replicate the benchmarks. These instructions serve as a guide for researchers, ensuring that they have the necessary information and resources at their disposal to validate and verify our findings.

We encourage researchers to leverage the resources available in the FinBen GitHub repository to replicate our benchmarks and explore further extensions and improvements. By promoting a culture of reproducibility, we aim to foster collaboration and drive advancements in the field of financial artificial intelligence.

## K    Response to Previous Review Concerns

In the previous submission to ACL ARR 2024 April, reviewers raised a few minor concerns about the justification of the datasets and the depth of insights derived. To address these, the following improvements have been made:

**Enhanced Dataset Justification:** The paper now includes a more detailed motivation for the inclusion of each dataset, highlighting their unique contributions and relevance to the financial sector.

**Incorporation of New Findings:** The paper presents new findings and insights derived from the comprehensive benchmark, showcasing how these results add to the current knowledge in the field of financial LLMs.

By addressing these concerns, the revised version of the paper aims to provide a more robust and insightful evaluation of LLMs in the financial sector while building on the previously recognized strengths.