# OpenReview forum: "FinBen: A Holistic Financial Benchmark for Large Language Models"
_NeurIPS.cc/2024/Datasets_and_Benchmarks_Track — NeurIPS 2024 Track Datasets and Benchmarks Poster_

### Official Review · Reviewer_m9nq · 2024-06-30
**Review of 1129**

**Rating:** 5
**Confidence:** 5
**Correctness:** Yes
**Clarity:** Yes

**Review:**

Cons:

1. The main work of this manuscript is to reorganize and classify existing datasets and does not contribute new datasets, which limits its contribution.

2. The FinBen 2.0 leaderboard in https://github.com/The-FinAI/PIXIU?tab=readme-ov-file is invalid. More representative LLMs should be included, such as Qwen series, XuanYuan series, etc.

3. There is a large blank space near Table 1 and Figure 1, which looks very abrupt.

4. The standard NeurIPS2024 format is not used, such as the lack of line numbers. The content after Checklist should be placed in supplementary materials, not the main body.

5. Typos: Nanjin Audit University


Questions:

1. There are already many financial evaluation datasets. This manuscript does not contribute a new dataset. So what is the main contribution?

2. Why not use the entire average or weighted average to compare the performance of the LLMs more intuitively?

**Strengths:**

1. The topic of evaluating LLMs in financial knowledge is important and interesting.

2. Comprehensive financial tasks are included in this benchmark.

**Additional Feedback:**

None

**Documentation:**

Yes

**Limitations:**

Yes

**Opportunities For Improvement:**

My main concern is the contribution of this work. Also, the cons and questions above should be addressed.

**Relation To Prior Work:**

Yes

**Summary And Contributions:**

This manuscript presents FinBen, a comprehensive benchmark specifically designed to evaluate LLMs in the English financial domain. It includes 36 diverse datasets spanning 24 tasks. 15 representative LLMs are evaluated to reveal their key advantages and limitations.

---

> ### Author Rebuttal · Authors · 2024-08-17
>
> Thank you for your thoughtful and detailed feedback. We sincerely appreciate the time and effort you’ve invested in reviewing our work and providing constructive suggestions. We are grateful for the recognition of our contributions and are committed to addressing the points you’ve raised to ensure our work is as impactful and clear as possible.
>
> **Main Contribution**
>
> Apologize for not sufficiently emphasizing our new datasets in the manuscript.  As detailed in Section 2.2, we also introduced new datasets that expand the scope and depth of evaluation in this emerging field.
> - FinTrade: A dataset featuring historical stock prices, news, and sentiment data for 10 stocks over one year, providing a unique challenge for agent-based financial trading scenarios.
> - Regulations: A long-form QA dataset focused on complex financial regulations within the European Union.
>
> For further details, please refer to the response to Reviewer M41M.
>
> Moreover, as highlighted in our general response and response to Reviewer wuHu, our contributions extend beyond reorganizing and classifying existing datasets. We have introduced new tasks, novel datasets, and innovative evaluation strategies. The core contribution of our work is the development of the most comprehensive open-source financial evaluation benchmark to date, examining the strengths and weaknesses of LLMs in the financial domain, to facilitate the advancement of financial LLMs and applications. The value and impact of our benchmark have been validated through the successful shared tasks at the FinNLP-AgentScen workshop during IJCAI-2024 and the ongoing shared task at the FinNLP workshop at COLING 2025. We will revise the introduction and method sections to better emphasize these contributions and ensure that the novel aspects of our work are clearly communicated.
>
> **Invalid leaderboard**
>
> We apologize for the misclarification regarding the leaderboard. Our leaderboard is available at https://huggingface.co/spaces/TheFinAI/Open-Financial-LLM-Leaderboard . Moreover, we have included more results on Spanish datasets as well as results of more representative LLMs including Qwen and Xuanyuan series. We are also committed to continually assessing additional LLMs and expanding the range of tasks covered by the leaderboard. Furthermore, we welcome the submission of new models for assessment to ensure that the benchmark remains comprehensive and up-to-date. We appreciate your feedback and are actively working to make the leaderboard as representative and inclusive as possible.
>
> **Formatting & Typos**
>
> We appreciate your feedback on the layout near Table 1 and Figure 1, and we will revise the formatting to ensure a more cohesive and visually balanced presentation. We will correct the typo for "Nanjin Audit University" to "Nanjing Audit University" in the manuscript. We’ll also make adjustments to align more closely with the NeurIPS 2024 formatting guidelines, including refining the placement of content and ensuring all required elements are properly formatted.
>
> **Average/weighted average for comparison**
>
> Thank you for your suggestion. As shown in our leaderboard (https://huggingface.co/spaces/TheFinAI/Open-Financial-LLM-Leaderboard), we now use the entire average to compare the performance of LLMs more intuitively. We will revise the relevant tables, figures, and discussions in our paper to reflect this approach and provide a clearer comparison of the models.

---

> > ### Author Rebuttal · Authors · 2024-08-29
> >
> > Building on your suggestions and to provide a more comprehensive evaluation, we have conducted additional assessments of the Qwen, Xuanyuan, and LLaMA 3.1 series across various tasks in the FinBen benchmark. The results are summarized below:
> >
> > | **Dataset**  | **Metrics** | **Qwen2-7B** | **Xuanyuan-6B** | **Qwen2-72B** | **Xuanyuan-70B** | **LLaMA3.1-8B** | **LLaMA3.1-70B** |
> > |--------------|-------------|--------------|-----------------|---------------|------------------|-----------------|------------------|
> > | **NER**      | Entity F1   | 0.07 | 0.06 | 0.02 | 0.08 | 0.14 | 0.05 |
> > | **FINER-ORD**| Entity F1   | 0.02 | 0.02 | 0.02 | 0.33 | 0.12 | 0.18 |
> > | **FinRED**   | F1          | 0.00 | 0.00 | 0.00 | 0.01 | 0.00 | 0.00 |
> > | **SC**       | F1          | 0.60 | 0.70 | 0.82 | 0.23 | 0.83 | 0.87 |
> > | **CD**       | F1          | 0.00 | 0.00 | 0.01 | 0.00 | 0.00 | 0.00 |
> > | **FNXL**     | Entity F1   | 0.00 | 0.00 | 0.00 | 0.00 | 0.00 | 0.00 |
> > | **FSRL**     | Entity F1   | 0.00 | 0.00 | 0.01 | 0.00 | 0.00 | 0.00 |
> > | **FPB**      | F1          | 0.52 | 0.74 | 0.75 | 0.52 | 0.76 | 0.79 |
> > |              | Acc         | 0.52 | 0.75 | 0.74 | 0.55 | 0.75 | 0.79 |
> > | **FiQA-SA**  | F1          | 0.57 | 0.56 | 0.63 | 0.82 | 0.75 | 0.74 |
> > | **TSA**      | RMSE↓       | 0.43 | 0.33 | 0.30 | 0.54 | 0.17 | 0.42 |
> > | **Headlines**| Avg F1      | 0.60 | 0.65 | 0.60 | 0.73 | 0.60 | 0.60 |
> > | **FOMC**     | F1          | 0.63 | 0.45 | 0.65 | 0.60 | 0.48 | 0.64 |
> > |              | Acc         | 0.64 | 0.51 | 0.66 | 0.61 | 0.56 | 0.67 |
> > | **FinArg-ACC**| MicroF1    | 0.43 | 0.47 | 0.57 | 0.58 | 0.53 | 0.65 |
> > | **FinArg-ARC**| MicroF1    | 0.53 | 0.60 | 0.63 | 0.67 | 0.55 | 0.55 |
> > | **MultiFin** | MicroF1     | 0.39 | 0.54 | 0.55 | 0.63 | 0.62 | 0.69 |
> > | **MA**       | MicroF1     | 0.83 | 0.84 | 0.84 | 0.79 | 0.85 | 0.84 |
> > | **MLESG**    | MicroF1     | 0.34 | 0.24 | 0.43 | 0.26 | 0.31 | 0.44 |
> > | **FinQA**    | EmAcc       | 0.00 | 0.00 | 0.00 | 0.00 | 0.00 | 0.00 |
> > | **TATQA**    | EmAcc       | 0.00 | 0.11 | 0.00 | 0.02 | 0.04 | 0.44 |
> > | **Regulations**| Rouge-1   | 0.31 | 0.24 | 0.31 | 0.30 | 0.27 | 0.10 |
> > |              | BertScore   | 0.68 | 0.64 | 0.69 | 0.67 | 0.65 | 0.61 |
> > | **ConvFinQA**| EmAcc       | 0.00 | 0.00 | 0.01 | 0.00 | 0.00 | 0.00 |
> > | **EDTSUM**   | Rouge-1     | 0.22 | 0.24 | 0.22 | 0.25 | 0.20 | 0.18 |
> > |              | BertScore   | 0.67 | 0.66 | 0.67 | 0.68 | 0.64 | 0.63 |
> > | **ECTSUM**   | Rouge-1     | 0.00 | 0.00 | 0.00 | 0.00 | 0.00 | 0.00 |
> > |              | BertScore   | 0.00 | 0.00 | 0.00 | 0.00 | 0.00 | 0.00 |
> > | **BigData22**| Acc         | 0.55 | 0.50 | 0.56 | 0.56 | 0.54 | 0.45 |
> > |              | MCC         | 0.01 | 0.00 | 0.05 | 0.06 | 0.03 | -0.00 |
> > | **ACL18**    | Acc         | 0.50 | 0.51 | 0.50 | 0.49 | 0.52 | 0.49 |
> > |              | MCC         | -0.02| 0.01 | -0.01| -0.02| 0.05 | 0.03 |
> > | **CIKM18**   | Acc         | 0.52 | 0.50 | 0.55 | 0.50 | 0.57 | 0.44 |
> > |              | MCC         | -0.03| -0.03| -0.01| 0.00 | -0.01| 0.02 |
> >
> > We will append these results and more discussions on our experimental findings in the final version of our paper. These results demonstrate our commitment to continually assessing a broader range of models to ensure our benchmark remains comprehensive and reflective of the latest advancements in financial LLMs. We are actively working on evaluating additional models and expanding the range of tasks covered by our leaderboard, with the goal of providing the most inclusive and up-to-date resource for the community.

---

> > > ### Author Response · Authors · 2024-08-31
> > > **Gentle Reminder Regarding the Discussion Period**
> > >
> > > Dear Reviewer m9nq,
> > >
> > > Thank you for reviewing our paper and providing valuable feedback. We would like to gently remind you that the discussion period will conclude on August 31.
> > >
> > > We have provided detailed and comprehensive responses to all your concerns and believe that we have thoroughly addressed your comments and clarified the questions raised. If you find that our responses have satisfactorily addressed your concerns, we would greatly appreciate it if you could consider adjusting your rating. Should you have any remaining questions or require further clarification, please don't hesitate to reach out to us. We are more than happy to continue the discussion.
> > >
> > > Thank you again for your time and constructive insights.

---

### Official Review · Reviewer_M41M · 2024-07-04
**Nice work**

**Rating:** 9
**Confidence:** 4
**Correctness:** N.A.
**Clarity:** I can nearly understand all the parts…

**Review:**

**Quality**: This paper provides a comprehensive evaluation of leveraging large language models for financial applications. A considerable number of datasets and tasks are utilized and numerous LLMs are adopted. Detailed discussions expound on the pros and cons of LLMs in finance. The overall quality of the work is awesome.

**Clarity**: The paper is easy to follow. However, given the massive contents of the dataset/task descriptions, the organization is a little bit scattered. It is not easy to remember all the details.

**Originality**: The originality lies in that it is the first work to integrate abundant financial datasets and tasks into a single benchmark. The authors further test the performances of SOTA LLMs on these datasets and tasks.

**Significance of this work**: I think this work provides insightful findings for financial LLM applications. Especially, the weakness of LLMs in certain applications (e.g., stock forecasting) is highlighted, which points out potential research directions for future endeavors.

**Strengths:**

1.	Well paper writing
2.	Sufficient amount of tasks/datasets/baselines
3.	Very comprehensive evaluations and discussions

**Additional Feedback:**

N.A.

**Documentation:**

The paper provides extensive detail on data collection and organization, availability, and maintenance, ensuring comprehensive documentation and intended uses. It also includes a URL for reviewer access, supporting reproducibility through thorough descriptions and released resources.

**Ethics:**

No concerns.

**Limitations:**

N.A.

**Opportunities For Improvement:**

If I have to point out some opportunities, I think the paper can be further organized. Though there are many details to present, it is never enough to think about how to coordinate them well.

Regarding the new datasets proposed by the authors, I think more descriptions are needed. Now, given Sec 2.2, it is still less clear for me to figure out where the data is collected from, and what is the frequency of data for FinTrade. Is it collected daily, intraday, or monthly….

**Relation To Prior Work:**

FinBen offers broader task coverage compared to prior work. It covers stock trading, novel agent-based and Retrieval-Augmented Generation (RAG) evaluation strategies, and introduces three novel open-source datasets.

**Summary And Contributions:**

The paper presents FinBen, an open-source benchmark for evaluating LLMs in finance, covering 36 datasets and 24 tasks. Key innovations include stock trading evaluation and new datasets for summarization, QA, and trading. Evaluations of 15 LLMs show strengths in information extraction but struggle with complex tasks. FinBen’s use in a shared task at FinNLP-AgentScen demonstrated its potential to advance financial LLMs.

---

> ### Author Rebuttal · Authors · 2024-08-17
>
> Thank you for your thorough and positive review of our work. We are grateful for your recognition of the quality, clarity, and significance of our contributions. We appreciate your suggestions for improvement.
>
> **Paper organization**
>
> We appreciate your suggestion regarding the organization of our paper. We understand the importance of a clear and coherent presentation, especially given the breadth of details involved. In response, we will continue to refine and reorganize our paper to enhance clarity and structure. This includes improving the organization of tasks, and results, adding more details of task taxonomy, more descriptions of data building and each evaluated task, and in-depth discussions of model performance, to ensure that our contributions are communicated as effectively as possible.
>
> **Dataset descriptions**
>
> We appreciate the opportunity to clarify these aspects. For each dataset, we will provide detailed information such as sources and collection methods et al, in the camera-ready version of our manuscript.
>
> For the FinTrade dataset, it includes the historical stock prices, fillings data and news data for 10 stocks over one year. (1) **Stock Price Data:** Each stock’s price data across 497 trading days is obtained from Yahoo Finance via the yfinance API. we obtained the OHLCV (open, high, low, close, adjusted close price, and volume) from Yahoo Finance at a *daily frequency*. The trading agent primarily references the adjusted close price to maintain consistency in the return series and to avoid the impact of corporate actions such as dividends, splits, etc. (2) **Filings Data:** We utilized the summary sections of Form 10-Q and Form 10-K data, which are publicly available from the EDGAR database of the U.S. Securities and Exchange Commission (SEC). The Form 10-Q is commonly referred to as the "quarterly report" while the Form 10-K is known as the "annual report." Over the course of a year, three "quarterly reports" and one "annual report" make the filings data available to the trading agent on a *quarterly basis*. (3) **News Data:** The news data aims to provide the agent a short-term view on the market. The news data is compiled with news titles and articles integrated from multiple publicly accessible news datasets such as [1] and acquired in a *daily frequency*. The data statistics of the data is shown in the following table:
>
>   | Ticker | Number of News | Number of 10-K/10-Q Files | Numerical Price Data |
>   |--------|----------------|---------------------------|----------------------|
>   | TSLA   | 3233           | 8                         | 497                  |
>   | NFLX   | 965            | 8                         | 497                  |
>   | AMZN   | 1675           | 8                         | 497                  |
>   | MSFT   | 1362           | 8                         | 497                  |
>   | AAPL   | 2082           | 8                         | 497                  |
>   | GOOG   | 1144           | 7                         | 497                  |
>   | DIS    | 1445           | 9                         | 497                  |
>   | GM     | 2252           | 9                         | 497                  |
>   | NIO    | 957            | 0                         | 497                  |
>   | COIN   | 1022           | 0                         | 497                  |
>
> For the Regulations dataset, the QA dataset is derived from the European Securities and Markets Authority's (ESMA) comprehensive document on the implementation of Regulation (EU) No 648/2012, known as EMIR, which pertains to over-the-counter (OTC) derivatives, central counterparties, and trade repositories. This dataset is an essential resource for understanding the regulatory framework that governs OTC derivatives in the European Union. EMIR, established in 2012, was designed to increase transparency and reduce the risks associated with derivatives trading. The document includes a series of questions and answers that clarify the application of EMIR, addressing various issues such as the obligations of financial and non-financial counterparties, the reporting requirements to trade repositories, and the procedures for calculating clearing thresholds. These Q&As are continuously updated to reflect the latest regulatory changes and interpretations, ensuring that market participants and authorities have the most current guidance. Our dataset comprises 254 QA pairs meticulously curated under the guidance of domain experts in financial regulation. These pairs were selected through a detailed analysis of the ESMA document to ensure relevance and accuracy, particularly in the practical application of EMIR. This makes the dataset an invaluable tool for both regulatory compliance and academic research in financial regulation, structured to facilitate easy access to specific queries and enhance understanding of the complex EMIR framework.
>
>
> [1] Dong, Zihan, Xinyu Fan, and Zhiyuan Peng. "FNSPID: A Comprehensive Financial News Dataset in Time Series." arXiv preprint arXiv:2402.06698 (2024).

---

> > ### Comment · Reviewer_M41M · 2024-08-22
> > **Thanks for your response**
> >
> > Thanks for your response. I maintain my positive view of the paper.

---

> > > ### Author Response · Authors · 2024-08-29
> > >
> > > Thank you for your positive feedback and for acknowledging our contributions. We are glad that our work has been well received and appreciate your support. Your insights have been invaluable in refining our analysis, and we are committed to ensuring that our final submission reflects the highest standards of research. We look forward to continuing to contribute to the advancement of financial large language models and benchmarks.

---

### Official Review · Reviewer_wuHu · 2024-07-23
**Review #1129**

**Rating:** 5
**Confidence:** 4
**Correctness:** Yes
**Clarity:** Good survey

**Review:**

Quality and Clarity: It is more like a survey paper that gathers all existing datasets and surveys the performance of different models. However, the difference between stock movement prediction tasks and stock trading tasks is not clear. That leads to the following question: What's the main difference between this work and last year's work?

Originality and Significance: The authors contribute to collecting and gathering all existing datasets together, but no contributions from the originality aspect, including both dataset and method parts.

The expectation of this track should be calling for a new dataset/benchmark, which shares new real-world tasks to challenge models. As I mentioned, this paper is more like a survey paper and could be submitted to a journal or a survey track.

**Strengths:**

(1) Collect many existing datasets together

(2) Explore many LLMs

**Additional Feedback:**

N/A

**Documentation:**

Provided.

**Opportunities For Improvement:**

I suggest writing more details and comparing the differences among datasets to submit as a survey paper. For example, markets and data collection periods could be provided. Additionally, some supervised methods (or state-of-the-art) for each task should be provided to compare the performance between LLMs and other models.

**Relation To Prior Work:**

It is not clear.

**Summary And Contributions:**

This paper adds one more decision-making task to last year's dataset track paper (https://neurips.cc/virtual/2023/poster/73431) and also enlarges the size of the dataset collection. Many models are explored.

---

> ### Author Rebuttal · Authors · 2024-08-17
>
> Thank you for your detailed feedback. We appreciate the opportunity to clarify our contributions. Our paper does provide a comprehensive overview of existing datasets and models. We also have made non-trivial contributions in introducing new tasks, novel datasets, and innovative evaluation strategies.
>
> Our work contributes in several key areas:
>
> 1. **Expanding Tasks and Datasets**: As shown in Table 1, FinBen introduces a substantially larger number of tasks and datasets than any previous benchmark, making it the most comprehensive and holistic open-source benchmark for financial LLMs to date (35 datasets and 24 tasks vs. 15 datasets and 8 tasks covered by the previous work PIXIU). Our benchmark uniquely covers seven critical aspects of the financial sector, including the evaluation of the stock trading task—a highly complex and critical area not addressed by any existing benchmarks. This task requires LLMs to make dynamic decisions on where to trade, at what price, and in what quantity, over a stochastic and complex stock market. This is fundamentally different from the stock movement prediction task, which only involves predicting stock movement based on historical data.
>
> 2. **Innovative Evaluation Strategies**: We introduce agent based evaluation in the stock trading task, which are new to the domain of financial LLMs. These strategies provide a more dynamic and realistic assessment of LLM capabilities, reflecting real-world applications where models must interact with and retrieve relevant information from vast datasets.
>
> 3. **Novel Datasets**: FinBen contributes several new open-source datasets that fill gaps in existing benchmarks and offer unique challenges for LLM evaluation, including:
>    - **FinTrade**: A dataset specifically for stock trading tasks, incorporating historical stock prices, news, and sentiment data for 10 stocks over one year.
>    - **Regulations**: A long-form QA dataset focused on Over-the-Counter (OTC) derivatives and financial regulations within the European Union.
>
>    More details of these datasets can be found in the response to Reviewer M41M
>
> 4. **Impact and Alignment with Track Objectives**: FinBen is aligned with the objectives of the dataset and benchmark track, as we contribute a novel holistic open-source benchmark with diverse challenging real-world financial tasks and novel datasets. Our benchmark has already proven its value by supporting and hosting the first shared task on financial LLMs at the FinNLP-AgentScen workshop during IJCAI-2024. This event attracted 35 registrations, demonstrating FinBen’s role in advancing the field by providing diverse, real-world challenges that push the boundaries of financial LLM development. Additionally, FinBen continues to support the advancement of financial LLMs by serving as the foundation for the shared task at the upcoming FinNLP workshop at COLING 2025 ([https://sites.google.com/nlg.csie.ntu.edu.tw/finnlp-fnp-llmfinlegal/home](https://sites.google.com/nlg.csie.ntu.edu.tw/finnlp-fnp-llmfinlegal/home)).
>
> In summary, we believe our paper is aligned with the track’s expectations and contributes the current most comprehensive open-source financial evaluation benchmark, incorporating diverse new challenging tasks, contributing novel datasets and evaluation strategies, and directly supporting the advancement of financial LLMs.

---

> > ### Author Rebuttal · Authors · 2024-08-29
> >
> > To further address the concerns regarding the comparison between LLMs and traditional non-LLM methods, we have included additional analysis in Appendix G, Table 14, where we compare the performance of LLMs against traditional methods on the stock movement prediction task. The table compares accuracy (ACC) and the Matthews correlation coefficient (MCC), especially in scenarios involving imbalanced datasets.
> >
> > As shown in Table 14, while LLMs such as GPT-4 and Gemini achieve competitive accuracy on stock movement prediction tasks (e.g., GPT-4 reaches 0.57 ACC on CIKM18), their MCC scores often lag behind those of traditional non-LLM models like SLOT. For instance, SLOT achieves 0.10 on BIGDATA22, 0.21 on ACL18, and 0.09 on CIKM18, compared to GPT-4’s 0.03, 0.02, and 0.02 on these datasets, respectively. This discrepancy highlights the potential limitations of LLMs in handling imbalanced datasets, where traditional models demonstrate a more balanced prediction performance.
> >
> > These results suggest that, although LLMs can achieve comparable accuracy, their lower MCC scores indicate a tendency to produce imbalanced predictions, revealing potential shortcomings in their predictive capabilities.
> >
> > Additionally, we conducted further comparisons of LLMs with classic non-LLM methods on risk management tasks. Our results indicate that even top-performing LLMs, such as Gemini, show a significant performance gap compared to non-LLM models. For example, in the Polish dataset, LLMs achieved an F1 score of 0.86 and an MCC of 0.14, while traditional methods reached a 0.99 F1 score and 0.57 MCC. This gap may stem from LLMs' weaker ability to process tabular data when it is converted to text, which poses challenges for effective feature inference—a domain where non-LLM models, specifically designed for tabular data, excel.
> >
> > The following table provides a summary of the performance comparison on risk management tasks:
> >
> > | **Dataset**         | **Method**            | **Metric** | **Value** |
> > |---------------------|-----------------------|------------|-----------|
> > | **Credit Card Fraud**| ANN [1]               | F1         | 0.85      |
> > |                     |                       | MCC        | 0.17      |
> > | **ccfraud**         | EGRNN++ [2]           | F1         | 0.90      |
> > |                     |                       | MCC        | 0.34      |
> > | **Polish**          | Bayesian [3]          | F1         | 0.99      |
> > |                     |                       | MCC        | 0.57      |
> > | **Travel Insurance**| Random Forest [4]     | F1         | 0.91      |
> > |                     |                       | MCC        | 0.15      |
> >
> > We also have expanded our analysis to include a comparison between traditional models and LLMs on various financial datasets. We will detail these comparisons in our appendix, particularly focusing on FPB, Headline, and NER. Below is a summary of the performance of traditional models from [5]:
> >
> > | **Model**          | **FPB (Acc)** | **NER (F1)** | **Headline (F1)** |
> > |--------------------|---------------|--------------|-------------------|
> > | **BERT-base**      | 0.856         | 0.79         | 0.967             |
> > | **FinBERT**        | 0.872         | 0.80         | 0.968             |
> > | **FLANG-BERT**     | 0.912         | 0.83         | 0.972             |
> > | **ELECTRA**        | 0.881         | 0.78         | 0.966             |
> > | **FLANG-ELECTRA**  | 0.919         | 0.82         | 0.980             |
> >
> > As shown in Table 3 of our paper, while LLMs demonstrate promising performance across various tasks, our comparison with traditional models underscores the strengths and potential gaps in current LLM capabilities. For instance, FLANG-ELECTRA achieves the highest accuracy of 0.919 on FPB, outperforming most LLMs, which typically range from 0.78 to 0.88 in accuracy. In NER, traditional models consistently outperform LLMs, with FLANG-BERT and FLANG-ELECTRA achieving F1 scores of 0.83 and 0.82, respectively, compared to LLMs like GPT-4, which matches but does not exceed these scores. Similarly, in Headline classification, FLANG-ELECTRA's F1 score of 0.980 surpasses that of even the best-performing LLMs. These results indicate that while LLMs offer strong generalization and versatility, traditional models continue to excel in tasks requiring high precision and deep contextual understanding, particularly in domain-specific financial tasks.
> >
> > These insights will be incorporated into our final manuscript, along with additional discussions to provide a deeper analysis of the limitations and potential improvements for LLMs in financial tasks. We hope these additions will address the concerns raised and underscore the significance of our contributions.
> >
> > **References:**
> >
> > [1] Asha, R. B., and Suresh Kumar KR. "Credit card fraud detection using artificial neural network." 2021.
> >
> > [2] Kamaruddin, Sk, and Vadlamani Ravi. "EGRNN++ and PNN++: Parallel and distributed neural networks for big data regression and classification." 2021.
> >
> > [3] Mukeri, Amir, Habibullah Shaikh, and Dr. DP Gaikwad. "Financial Data Analysis Using Expert Bayesian Framework For Bankruptcy Prediction." 2020.
> >
> > [4] Li, Xiaonan. "Exploring the Potential of Machine Learning Techniques for Predicting Travel Insurance Claims: A Comparative Analysis of Four Models." 2023.
> >
> > [5] Shah, R. S., Yang, A., Chen, Z., Zhang, T., et al. (2022). When FLANG meets FLUE: Benchmarks and large pre-trained language models for financial domain. *arXiv preprint arXiv:2211.00083.*

---

> > > ### Author Response · Authors · 2024-08-31
> > > **Gentle Reminder Regarding the Discussion Period**
> > >
> > > Dear Reviewer wuHu,
> > >
> > > Thank you for reviewing our paper and providing valuable feedback. We would like to gently remind you that the discussion period will conclude on August 31.
> > >
> > > We have provided detailed and comprehensive responses to all your concerns and believe that we have thoroughly addressed your comments and clarified the questions raised. If you find that our responses have satisfactorily addressed your concerns, we would greatly appreciate it if you could consider adjusting your rating. Should you have any remaining questions or require further clarification, please don't hesitate to reach out to us. We are more than happy to continue the discussion.
> > >
> > > Thank you again for your time and constructive insights.

---

### Official Review · Reviewer_9EDL · 2024-07-24
**FinBen: An Holistic Financial Benchmark for Large Language Models**

**Rating:** 7
**Confidence:** 3
**Correctness:** The claims made in the submission are…
**Clarity:** Yes, the paper is well written.

**Review:**

The paper presents a comprehensive approach to evaluating LLMs in the financial domain, employing a systematic methodology and thorough analysis of 15 representative LLMs, with experimental results offering valuable insights and references. FinBen introduces a broader range of tasks and datasets compared to existing benchmarks, including evaluations of stock trading and novel agent-based and RAG evaluation strategies. The significance of this work lies in its potential to drive innovation and research in the field of financial LLMs.

**Strengths:**

The paper's strengths lie in its comprehensive and systematic approach to evaluating LLMs in the financial domain. FinBen covering a broader range of tasks and datasets, contributes to the field by addressing previously overlooked areas such as stock trading and evaluation strategies. The quality of the research is evident in the thorough analysis of 15 representative LLMs, providing valuable insights and highlighting both strengths and limitations.

**Additional Feedback:**

None

**Documentation:**

More detailed descriptions of the data collection and organizaition process should be provided.

**Ethics:**

There are no apparent ethical concerns with the submission.

**Limitations:**

The authors addressed some major limitations of their work

**Opportunities For Improvement:**

1. Expanding the dataset to include more diverse global financial markets and multiple languages would enhance FinBen's applicability and relevance in a broader context beyond the American market and English texts.

2. In some areas, such as forecasting and risk management results, the research is not sufficiently in-depth. There should be a quantitative comparison of the performance of LLMs with classic non-LLMs methods, followed by an analysis of the directions for improvement in LLMs.

**Relation To Prior Work:**

Yes, the paper clearly discusses how this work differs from previous contributions.

**Summary And Contributions:**

The paper introduces FinBen, an extensive open-source evaluation benchmark for LLMs in the financial domain. FinBen comprises 36 datasets spanning 24 tasks across seven critical aspects: information extraction, textual analysis, question answering, text generation, risk management, forecasting, and decision-making. Key contributions include a broader range of tasks and datasets than existing benchmarks, and three new open-source datasets. The systematic evaluation of 15 LLMs highlights their strengths and limitations, particularly their proficiency in information extraction and textual analysis, while identifying challenges in advanced reasoning and complex tasks. The experimental results provide a solid foundation and data environment for future research on financial LLMs.

---

> ### Author Rebuttal · Authors · 2024-08-17
>
> We highly appreciate your effort in raising additional questions that can further strengthen the work. We address the concerns about FinBen in our main response and provide clarifications on questions below.
>
> **Expanding Dataset**
>
> We agree that expanding the dataset to include more diverse global financial markets and multiple languages is crucial for enhancing FinBen’s applicability. We have already explored open-source multilingual financial evaluation benchmarks, focusing on Spanish [1] and Chinese [2]. In response, we will incorporate Spanish financial tasks from [1] into our evaluation and analyze the performance of LLMs in the camera-ready version of our manuscript. The dataset details and the updated results are presented in Table 1 and 2 of the attached rebuttal PDF, respectively.
>
> [1] Zhang, Xiao, et al. "Dólares or Dollars? Unraveling the Bilingual Prowess of Financial LLMs Between Spanish and English." KDD 2024
>
> [2] Hu, Gang, et al. "No Language is an Island: Unifying Chinese and English in Financial Large Language Models, Instruction Data, and Benchmarks." arXiv preprint arXiv:2403.06249 (2024).
>
> **Traditional Methods Comparison**
>
> Due to space limitations, we have already included the comparison between LLMs and traditional non-LLM methods on the forecasting task – the stock movement prediction – in Appendix G, Table 14. The table compares accuracy (ACC) and the Matthews correlation coefficient (MCC, particularly in scenarios with imbalanced datasets) across three datasets.
>
> As shown in Table 14, while some LLMs like GPT-4 and Gemini achieve competitive accuracy in stock movement prediction (e.g., GPT-4 reaches 0.57 ACC on CIKM18), their MCC often lags behind non-LLM models such as SLOT. For example, SLOT achieves 0.10 on BIGDATA22, 0.21 on ACL18, and 0.09 on CIKM18, compared to GPT-4’s 0.03, 0.02, and 0.02 on these datasets, respectively. This suggests that although LLMs can achieve comparable prediction accuracy, their lower MCC scores indicate a tendency to make imbalanced predictions, reflecting potential limitations in their predictive abilities.
>
> These discrepancies can be attributed to two primary limitations of current LLM models:
> 1. LLMs are restricted by limited context lengths, reducing their ability to encode and analyze long-term dependencies in stock historical data—an area where SOTA traditional non-LLM methods excel.
> 2. LLMs often extract shallow features, such as trends from price data, and struggle with complex numerical reasoning tasks essential for accurate financial forecasting.
>
> To address these limitations, future work could focus on enhancing LLMs’ context length capabilities, allowing them to process longer historical sequences. Additionally, improving LLMs' numerical reasoning abilities could lead to more accurate and robust predictions, potentially surpassing SOTA non-LLM models.
>
> Furthermore, we have included results comparing LLMs with classic non-LLM methods on risk management tasks. Even the top-performing LLMs, such as Gemini, show a significant performance gap compared to non-LLM models (e.g., 0.86 F1, 0.14 MCC vs. 0.99 F1, 0.57 MCC in Polish). This gap may be due to LLMs’ weaker ability to process tabular data when converted to text, leading to challenges in effective feature inference—an area where non-LLM models, specifically designed for tabular data, excel.
>
> We will incorporate these new results and in-depth discussions in the camera-ready version of our manuscript.
>
> ### Performance Comparison on Risk Management Tasks
>
> | Datasets              | Methods           | Metrics | Value |
> |-----------------------|-------------------|---------|-------|
> | Credit Card Fraud      | ANN [1]           | F1      | 0.85  |
> |                       |                   | MCC     | 0.17  |
> | ccfraud               | EGRNN++ [2]       | F1      | 0.90  |
> |                       |                   | MCC     | 0.34  |
> | Polish                | Bayesian [3]      | F1      | 0.99  |
> |                       |                   | MCC     | 0.57  |
> | Travel Insurance      | random forest [4] | F1      | 0.91  |
> |                       |                   | MCC     | 0.15  |
>
> [1] Asha, R. B., and Suresh Kumar KR. "Credit card fraud detection using artificial neural network." 2021.
>
> [2] Kamaruddin, Sk, and Vadlamani Ravi. "EGRNN++ and PNN++: Parallel and distributed neural networks for big data regression and classification." 2021.
>
> [3] Mukeri, Amir, Habibullah Shaikh, and Dr. DP Gaikwad. "Financial Data Analysis Using Expert Bayesian Framework For Bankruptcy Prediction." 2020.
>
> [4] Li, Xiaonan. "Exploring the Potential of Machine Learning Techniques for Predicting Travel Insurance Claims: A Comparative Analysis of Four Models." 2023.

---

> > ### Author Rebuttal · Authors · 2024-08-29
> >
> > **Additional Results for Expanding Dataset**
> >
> > The table below summarizes the performance of different LLMs across key Spanish datasets, providing a clear comparison with traditional models. Notably, models such as GPT-4 and Qwen2-7B-Instruct demonstrate strong performance across several tasks, outperforming others in areas like MultiFin-ES and TSA. However, there are areas where certain models, like XuanYuan 6B-Chat, excel, particularly in tasks such as EFP and FinanceES. This detailed comparison underscores the importance of continued expansion and evaluation of Spanish datasets to fully understand the capabilities and limitations of LLMs in the financial domain. These insights will be incorporated into our final submission to ensure that our benchmark remains comprehensive and representative of diverse linguistic and financial challenges.
> >
> > | **Dataset**        | **Metrics** | **ChatGPT** | **GPT-4** | **LLaMA2 7B-chat** | **LLaMA3 8B** | **FinMA 7B** | **InternLM 7B** | **Falcon 7B** | **Mixtral 7B** | **CFGPT sft-7B-Full** | **Qwen2 7B** | **XuanYuan 6B-Chat** |
> > |--------------------|-------------|-------------|-----------|--------------------|---------------|--------------|-----------------|---------------|----------------|-----------------------|--------------|-----------------------|
> > | **MultiFin-ES**    | ACC         | 0.48        | **0.60**  | 0.23               | 0.11          | 0.25         | 0.09            | 0.05          | 0.13           | 0.02                  | 0.43         | 0.30                  |
> > |                    | F1          | 0.47        | **0.60**  | 0.11               | 0.12          | 0.27         | 0.12            | 0.07          | 0.17           | 0.03                  | 0.42         | 0.27                  |
> > | **EFP**            | ACC         | 0.30        | 0.27      | 0.27               | 0.27          | 0.35         | 0.27            | 0.27          | 0.27           | 0.24                  | 0.27         | **0.41**               |
> > |                    | F1          | **0.47**    | 0.19      | 0.12               | 0.12          | 0.21         | 0.12            | 0.12          | 0.12           | 0.20                  | 0.14         | **0.41**               |
> > | **EFPA**           | ACC         | 0.31        | 0.34      | 0.26               | 0.20          | 0.35         | 0.25            | 0.26          | 0.25           | 0.23                  | 0.32         | **0.38**               |
> > |                    | F1          | 0.25        | 0.27      | 0.10               | 0.09          | 0.21         | 0.10            | 0.10          | 0.12           | 0.22                  | 0.18         | **0.37**               |
> > | **FinanceES**      | ACC         | 0.13        | 0.15      | 0.14               | 0.20          | 0.02         | 0.12            | 0.15          | 0.13           | 0.01                  | 0.05         | **0.30**               |
> > |                    | F1          | 0.08        | 0.09      | 0.13               | 0.23          | 0.03         | 0.16            | 0.18          | 0.20           | 0.02                  | 0.05         | **0.30**               |
> > | **TSA**            | ACC         | 0.21        | 0.47      | 0.07               | 0.03          | 0.04         | 0.02            | 0.06          | 0.001          | 0.02                  | 0.07         | **0.53**               |
> > |                    | F1          | 0.24        | 0.46      | 0.04               | 0.06          | 0.07         | 0.04            | 0.10          | 0.002          | 0.04                  | 0.05         | **0.52**               |
> > | **FNS**            | Rouge-1     | 0.02        | **0.19**  | 0.00               | 0.00          | 0.00         | 0.00            | 0.00          | 0.00           | 0.00                  | 0.05         | 0.02                  |
> > |                    | Rouge-2     | 0.04        | **0.06**  | 0.00               | 0.00          | 0.00         | 0.00            | 0.00          | 0.00           | 0.00                  | 0.05         | 0.00                  |
> > |                    | Rouge-L     | 0.12        | **0.13**  | 0.00               | 0.00          | 0.00         | 0.00            | 0.00          | 0.00           | 0.00                  | 0.05         | 0.02                  |

---

> > > ### Author Response · Authors · 2024-08-31
> > > **Gentle Reminder Regarding the Discussion Period**
> > >
> > > Dear Reviewer 9EDL,
> > >
> > > Thank you for reviewing our paper and providing valuable feedback. We would like to gently remind you that the discussion period will conclude on August 31.
> > >
> > > We have provided detailed and comprehensive responses to all your concerns and believe that we have thoroughly addressed your comments and clarified the questions raised. If you find that our responses have satisfactorily addressed your concerns, we would greatly appreciate it if you could consider adjusting your rating. Should you have any remaining questions or require further clarification, please don't hesitate to reach out to us. We are more than happy to continue the discussion.
> > >
> > > Thank you again for your time and constructive insights.

---

### Author Rebuttal · Authors · 2024-08-17

We sincerely thank the reviewers for their time and detailed feedback. To address the concerns raised:
1. We have added evaluation results on six Spanish tasks, comparisons between LLMs and classic non-LLM methods in risk management tasks, as well as results from more representative LLMs like Qwen and XuanYuan. We will continue to update results throughout the discussion period.
2. We have enhanced the descriptions of the proposed new datasets and provided in-depth discussions on forecasting and risk management results.
3. Our leaderboard is accessible at https://huggingface.co/spaces/TheFinAI/Open-Financial-LLM-Leaderboard, where we have used the overall average to intuitively compare LLM performance.

Moreover, we appreciate the opportunity to clarify our contributions:
1. FinBen is the most comprehensive and holistic open-source benchmark for financial LLMs to date, encompassing 35 datasets and 24 tasks across seven critical categories of financial NLP tasks and applications.
2. It is the first benchmark to include the evaluation of agent-based stock trading tasks, offering a dynamic and realistic assessment of LLM capabilities.
3. We contribute several new datasets to the research community, providing unique challenges and advancing the field.
4. We have systematically evaluated 15 LLMs using FinBen, thoroughly examining their strengths and weaknesses in the financial domain

The value and impact of our benchmark have been validated through the successful shared tasks at the FinNLP-AgentScen workshop during IJCAI-2024 and the ongoing shared task at the FinNLP workshop at COLING 2025 (https://sites.google.com/nlg.csie.ntu.edu.tw/finnlp-fnp-llmfinlegal/home ). In summary, we believe our paper aligns with the track’s expectations by contributing the most comprehensive open-source financial evaluation benchmark available, incorporating diverse and challenging tasks, novel datasets, and innovative evaluation strategies, all of which directly support the advancement of financial LLMs.

---

### Decision · Program_Chairs · 2024-09-26

**Decision:**

Accept (Poster)

**Comment:**

The paper introduces FinBen, which gathers a lot of existing finance benchmark datasets together and adds 3 new datasets,
regularization, EDTSum, finTrade.  While the number of new datasets is very limited, it is still valuable to put everything together in one place.  On the other hand, we hope the authors can seriously consider the concerns of our reviewers on the number of new datasets.  If possible, please add more to the proposed benchmark to demonstrate that "it is a new benchmark."   As the paper includes many different datasets and tasks, one reviewer suggests the authors to think about organizing the content in a better way. Please give this concern careful consideration during the revision process, as addressing it will greatly enhance the benchmark’s acceptance among readers.